# Antarctic Peninsula warm winters influenced by Tasman Sea temperatures

Kazutoshi Sato[1,2✉], Jun Inoue[3,4,2], Ian Simmonds [5] & Irina Rudeva [6,5]

The Antarctic Peninsula of West Antarctica was one of the most rapidly warming regions on the Earth during the second half of the 20th century. Changes in the atmospheric circulation associated with remote tropical climate variabilities have been considered as leading drivers of the change in surface conditions in the region. However, the impacts of climate variabilities over the mid-latitudes of the Southern Hemisphere on this Antarctic warming have yet to be quantified. Here, through observation analysis and model experiments, we reveal that increases in winter sea surface temperature (SST) in the Tasman Sea modify Southern Ocean storm tracks. This, in turn, induces warming over the Antarctic Peninsula via planetary waves triggered in the Tasman Sea. We show that atmospheric response to SST warming over the Tasman Sea, even in the absence of anomalous tropical SST forcing, deepens the Amundsen Sea Low, leading to warm advection over the Antarctic Peninsula.

[1] Kitami Institute of Technology, Kitami, Japan. [2] Application Laboratory, Japan Agency for Marine-Earth Science and Technology, Yokohama, Japan. [3] National Institute of Polar Research, Tachikawa, Japan. [4] The Graduate University for Advanced Studies, SOKENDAI, Hayama, Japan. [5] School of Earth Sciences, The University of Melbourne, Melbourne, Australia. [6] Australian Bureau of Meteorology, Melbourne, Australia. ✉email: satokazu@mail.kitami-it.ac.jp

The West Antarctic region, and the Antarctic Peninsula (AP) in particular, experienced dramatic temperature changes from the International Geophysical Year 1958 to the late 20th century. Significant long-period warming of the troposphere has been revealed in observations (e.g., from surface stations[1–3], radiosondes[4,5], and satellites[6,7]) and reanalysis data[8,9] in all seasons, with the highest warming rate in winter. Although positive trends in surface air temperature were found in the entire AP, the warming rate has been particularly marked at stations on the west side of the AP[2,8]. The AP air temperature trends are sensitive to the chosen start and end years of periods and period length[3,7,10,11]. There has been a statistically significant warming trend over the AP for a long period[1,3,4,6,7,9–11]. In contrast, for short periods, warming trends over the AP are not seen during austral summer, essentially because of natural internal variability[3,10,11].

The increases in air temperature over Antarctica are related to enhanced warm advection associated with changes in the key atmospheric circulations over the Southern Ocean (SO) (e.g., the Amundsen Sea Low [ASL][12–16] and the Southern Annular Mode [SAM][17–21]). The ASL, the sea level pressure (SLP) structure of which is related to the number and intensity of low-pressure systems over the Amundsen Sea, plays an important role in the West Antarctic surface climate (e.g., surface air temperature, sea ice extent, and wind speed), particularly during winter when cyclones are more numerous and deeper[14–16]. The SAM, characterized by westerly circumpolar flow variability associated with the strong meridional pressure gradient between the high and mid-latitudes of the Southern Hemisphere (SH), greatly influences synoptic-scale activity over the SO. A positive SAM causes poleward displacement of the cyclone tracks and reinforces the ASL, promoting surface warming over the AP[16,20,21].

The relationships between the atmospheric circulation over the SO and tropical oceanic variability, often called 'tropical-polar teleconnections', have been examined in previous studies[8,9,16,22–41]. In the austral cold seasons, heating by anomalous sea surface temperatures (SSTs) in the tropical region generates a Rossby wave train from the tropics to the Antarctic region via the SO, influencing its teleconections[8,9,22–24]. By this mechanism, the El Niño-Southern Oscillation (ENSO) modulates the position and strength of the ASL[16,22,23,25,26]. Numerical experiments have shown that warming over the tropical Atlantic and Indian Oceans and cooling over the tropical eastern Pacific Ocean result in a deeper ASL and West Antarctic warming[23]. In addition, the longitudinal Indian Ocean SST contrast, known as the Indian Ocean Dipole (IOD), has a remote effect on the climate over the SH[40,41]. When the eastern Indian Ocean has a cold SST anomaly compared with the western Indian Ocean (positive phase of the IOD) without ENSO, the pressure anomaly is negative (positive) north (south) of the Ross Sea (Australia), promoting sea ice formation west of the Ross Sea[41]. However, a recent study has revealed considerable uncertainties in both the pattern and amplitude of the atmospheric response to ENSO due to interannual variability in the extratropics[42]. Various studies have reported strong Northern Hemisphere high-latitude atmospheric responses to atmospheric and oceanic forcing over the northern mid-latitudes[43–48]. However, no previous study has reported the impact of change in SSTs in the SH mid-latitudes on Antarctic climate variability.

In this study, we investigate the linkage between ocean variabilities in the Southern Hemisphere mid-latitudes and Antarctic warming using reanalysis datasets and an atmospheric general circulation model. We reveal that warming in the Tasman Sea strengthens the meridional SST gradient between mid-latitude and high-latitude in the Southern Hemisphere, promoting AP warming through the poleward shift of storm tracks over the SO. In addition, our model experiments show that the increase in SST in the Tasman Sea alone produces warming in the AP even without anomalous tropical SST cooling.

## Results

### Warm and cold winters at six stations on the Antarctic Peninsula.

To understand the connection between interannual variability of SSTs in the mid-latitudes and changes in AP air temperature, we evaluated a time series of averaged surface air temperature anomalies for winter (June to August) and the other three seasons over the period 1979–2019 at six stations in the AP (Fig. 1a, see "Methods" section). The data exhibited an inter-annual half standard deviation (0.5σ) of 1.0 °C. The averaged temperatures anomalies in thirteen years exceeded +0.5σ ('warm' AP winters), whereas twelve winters recorded mean temperatures less than −0.5σ ('cold' AP winters). Although weak negative trends of AP air temperature since 1999 were reported by a previous study[10], our analysis shows that the AP stations experienced eight warm winters since that time (in 2000, 2004, 2008, 2010, 2014, 2016, 2018, and 2019). To investigate the causes of the temperature differences, we constructed difference maps of atmospheric and oceanic fields between composites of warm and cold AP winters using the Climate Forecast System Reanalysis (CFSR) provided by the National Centers for Environmental Prediction[49,50] (see "Methods" section), which have relatively small biases in atmospheric parameters[51]. A negative sea ice concentration anomaly is seen near the AP, suggesting that this sea ice loss contributed to and/or resulted from the warm temperature (Fig. 1b). However, an east-west seesaw pattern of high and low sea ice concentration anomalies was observed between the Amundsen Sea and the area near the AP, resulting from sea ice drifting speed anomalies near the AP and over the Amundsen Sea. Therefore, surface conditions (e.g., air temperature and sea

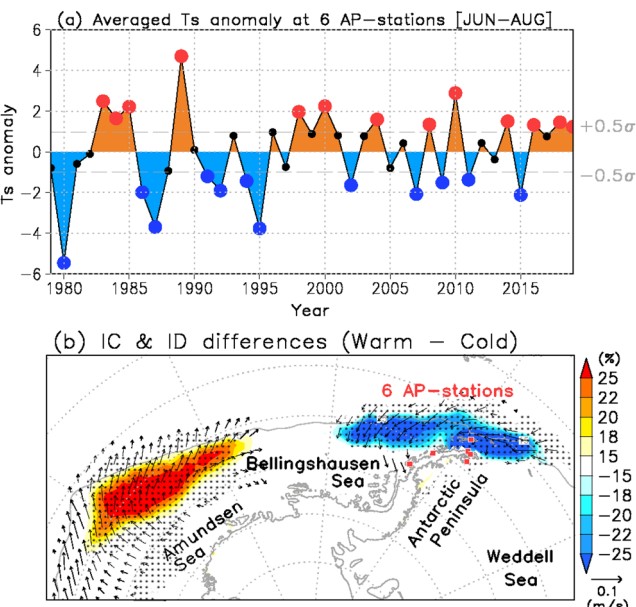

**Fig. 1 Surface anomalies of West Antarctica. a** Time series of averaged surface air temperature anomalies (°C, deviation from climatology for 1979–2019) during June to August at six Antarctic Peninsula stations (Bellingshausen, O'Higgins, Esperanza, Marambio, Vernadsky and Rothera). Red and blue dots indicate the top thirteen warm winters and twelve cold winters. Dashed lines show one-half standard deviation (±0.5σ). **b** Difference map of sea ice concentration (IC) (%: shaded) and ice drifting speed (ID) (m/s: vector) between warm and cold winters. Dotted areas denote significant differences exceeding the 95% confidence level.

ice concentration) in these regions would be induced by atmospheric circulation anomalies over the SO.

**Atmospheric circulation and Antarctic Peninsula warming.** Figure 2a shows the differences in temperature at 700 hPa (T700) and SLP between warm and cold AP winters. There is a cyclonic anomaly over the Amundsen Sea and an anti-cyclonic anomaly off the east coast of Argentina. This pattern resulted in a strong SLP gradient over the Drake Passage, leading to northerly warm advection over the Weddell Sea and the AP. The poleward drifting of sea ice associated with the northerly wind anomaly decreased sea ice extents over the Bellingshausen Sea and Drake Passage (Fig. 1b). The strong Amundsen Sea cyclonic anomaly, which has also been referred to as the ASL in some studies[12,14,15,18], enhanced warm advection over the AP. In contrast, the southerly cold advection from Antarctica induced cold temperature anomalies and an equatorward shift of sea ice over the Ross and Amundsen Seas.

The 300 hPa geopotential height (Z300) anomalies show a strong wave-like structure excited over the SO (Fig. 2b), resembling a zonal wave 3 pattern[52]. This pattern is similar to atmospheric responses to tropical SST anomalies reported by previous studies (e.g., tropical-polar teleconnections[8,9,16,22–41]), although the strongest wave anomaly pattern appears over the mid-latitudes and high latitudes. The propagation of wave activity flux as indicated by arrows[53] shows a wave train originating from the south of the New Zealand region. This suggests that these wave-like anomaly patterns are associated with the variability of the mid-latitudes, particularly changes in SST in the Tasman Sea because there was a positive relationship between SST in the Tasman Sea and T700 over the AP (Supplementary Fig. 1).

To address the causes of SLP anomalies over the SO and Antarctica, we show composite maps of the densities of cyclones in warm and cold AP winters (see "Methods" section). An area of high cyclone density was found around the Antarctic coastline in both composites (Supplementary Fig. 2). These patterns are consistent with the results of previous studies[12–15,18,21]. However, there were positive differences in the density of cyclones over the Antarctic coastal regions, particularly in the Pacific sector (Fig. 2c). In contrast, negative differences were seen over the Drake Passage and in the central Pacific to the east of New Zealand. In addition, positive precipitation anomalies over the south-eastern Pacific and negative anomalies to the east of Argentina and New Zealand are consistent with the geographical distribution of changes in cyclone density (Fig. 2c, d). These patterns suggest that the cyclones tended to shift poleward in warm AP winters.

**Impact of Tasman Sea warming on the atmospheric circulation over the Southern Ocean.** In recent years, considerable warming has occurred in the Tasman Sea[54], and the frequency and duration of marine heatwaves have increased[55,56]. Satellite data show a change in the South Pacific subtropical gyre associated with enhanced wind stress curl leading to recent SST warming in the Tasman Sea in all seasons[57]. In addition, the warm East Australian Current in the west Tasman Sea has exhibited a poleward shift and intensification in recent times[58]. Mid-latitude and polar teleconnection patterns are seen between the Tasman Sea and Antarctica[59]. A warming Tasman Sea has strengthened the westerlies between high and mid-latitudes[60] (increased baroclinicity) and thus has influenced cyclone tracks over the SO. Therefore, these atmospheric circulation changes related to anomalous SST warming over the Tasman Sea would contribute to recent anomalous warm AP winters.

To investigate SST anomalies in the Tasman Sea, we focus on the difference in SST between warm and cold AP winters (Fig. 2e, Supplementary Fig. 3a). A remarkable feature of Fig. 2e is that the SST differences over most of the SO are quite small, with the notable exception of the significant SST differences in the Tasman Sea and to the east of New Zealand. The zonal wind speed at 300 hPa has its strongest positive anomaly to the south of New Zealand (Fig. 2f), similar to split jet patterns with strong polar jets[27,61]. However, in warm AP winters, the polar jet tends to shift poleward over the region south of Australia and New Zealand (Supplementary Figs. 3b, 4). In warm AP winters, the higher SST in the Tasman Sea strengthens the meridional SST gradient between the Tasman Sea and the SO (Supplementary Fig. 3c). Therefore, the poleward shift of the upper-level jet south of the Tasman Sea would be induced by not only La Niña and a positive SAM but also by this increase in this meridional SST gradient (Fig. 2f, Supplementary Fig. 4). The upper-level jet anomaly shifts cyclone tracks to the south over the Tasman Sea, causing the change in the atmospheric heating in the downstream regions (i.e., the Pacific sector of the SO).

**Antarctic Peninsula warming without tropical heating effects.** Previous investigations have shown that strengthening of the ASL can be induced by La Niña, a positive SAM, and a negative IOD[8,9,16,22–41]. In warm AP winters, the wave activity flux anomalies appear to originate from the subtropical Pacific to Antarctica (Fig. 3a). The tropical Pacific has a negative SST anomaly (Fig. 3b) that includes the effect of La Niña and would influence atmospheric circulation over the SH, as reported by previous studies[8,9,16,22–39]. To investigate the atmospheric response to tropical SST anomalies in strong ENSO years, we composited the atmospheric and ocean fields for strong La Niña and El Niño winters (see "Methods" section, Fig. 3c, d, and Supplementary Figs. 5a, 6b). In the strong La Niña winters, although the wave-like anomaly patterns are similar to those in warm AP winters, a significant wave train appears to originate from the tropical Pacific region and extend to West Antarctica (Fig. 3a, c). In addition, the amplitudes and areas of tropical Pacific SST cooling are greater than those in warm AP winters (Fig. 3b, d). To remove the impact of strong ENSO in warm and cold AP winters, we determined the differences in atmospheric and ocean fields between warm and cold AP winters without strong ENSO winters (i.e., strong La Niña [1984, 1985, 1989, 1998, 2000, 2010] and strong El Niño [1987, 1991, 2002, 2009, 2015] winters) (see "Methods" section, Fig. 3e, f, and Supplementary Figs. 5b, 6c). In warm AP winters without strong La Niña, the SST cooling anomaly associated with La Niña events is not seen over the central Pacific Ocean (Fig. 3f). However, these winters exhibit a positive SST anomaly over the Tasman Sea and a significant wave-like anomaly pattern from the subtropical Pacific to Antarctica, suggesting that Tasman Sea warming anomalies contributed to these wave-like anomalies from the subtropical Pacific to West Antarctica.

We conducted similar analyses for strong SAM and strong IOD winters (see "Methods" section, Supplementary Figs. 5, 6, and 7). In strong positive SAM winters, the atmospheric circulation anomalies resemble those in warm winters over the AP (Supplementary Figs. 6d, 7a, b). The northeast AP warming, which has a significant positive correlation with the SAM[33], is seen in strong positive SAM winters (Supplementary Fig. 6d). However, over the Pacific sector of the SO, the amplitudes of atmospheric circulation and wave activity flux anomalies in strong positive SAM winters are smaller than those in warm AP winters (Fig. 3a and Supplementary Fig. 7a). In addition, in warm AP winters without a strong SAM, the wave activity flux from

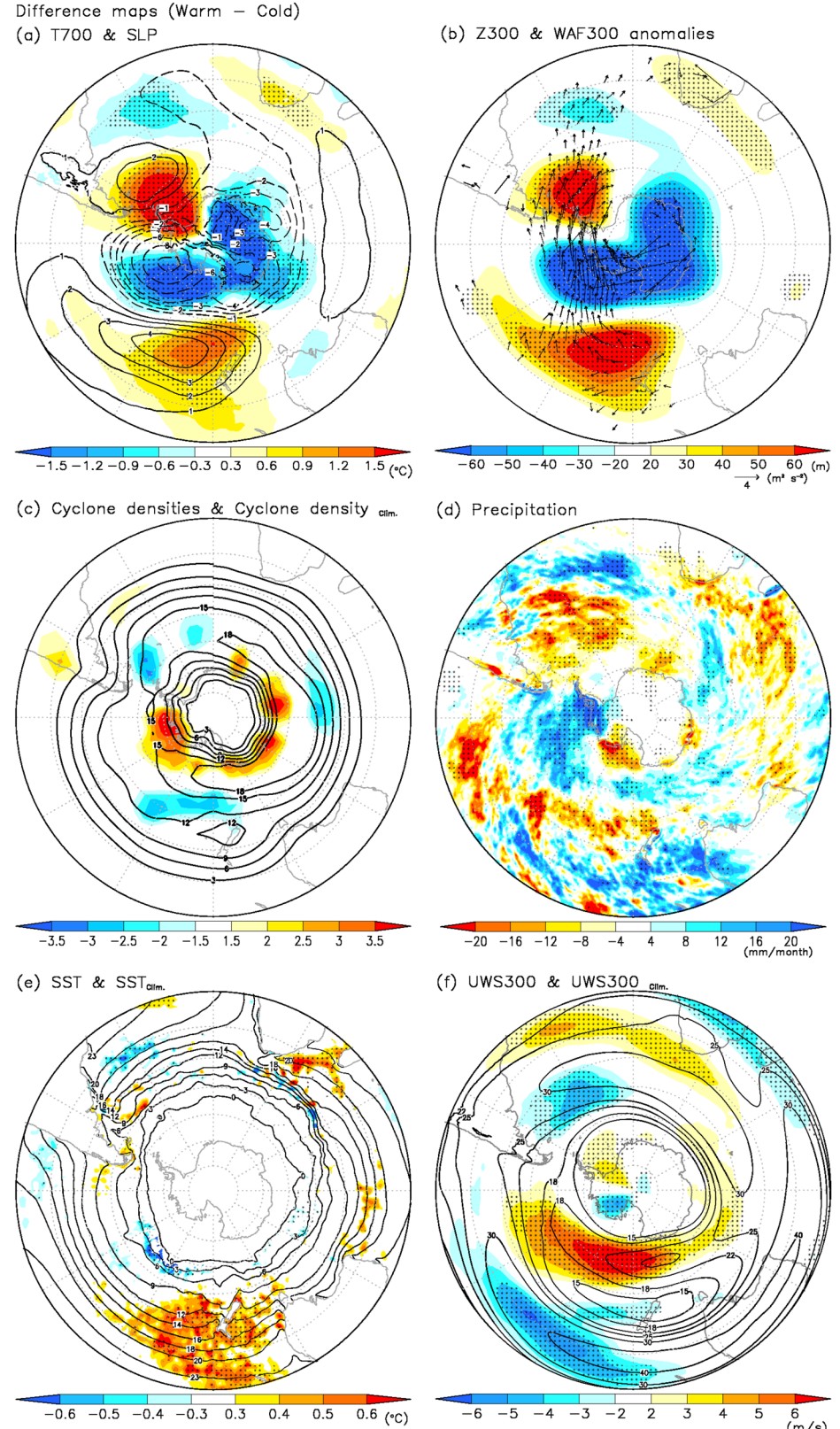

**Fig. 2 Atmospheric circulation and oceanic anomalies between warm and cold AP winters over the Southern Hemisphere.** Difference maps for (**a**) air temperature at 700 hPa (T700) (°C: shaded) and sea level pressure (SLP) (hPa: contours), (**b**) geopotential height at 300 hPa (Z300) (m: shaded) with the horizontal component of wave-activity flux anomalies (m²/s²: vector) at 300 hPa (WAF300) from ref. [53], (**c**) cyclone density (shaded: count/S) with climatological cyclone density (count/S: contours), (**d**) precipitation (mm/month), (**e**) sea surface temperatures (SSTs) between warm and cold winters (°C: shaded) with climatological SST (°C: contours) and (**f**) U-wind speed at 300 hPa (UWS300) (m/s: shaded) with climatological UWS300 (m/s: contours) for 1979–2019 between warm and cold AP winters. Dotted areas denote significant differences exceeding the 95% confidence level.

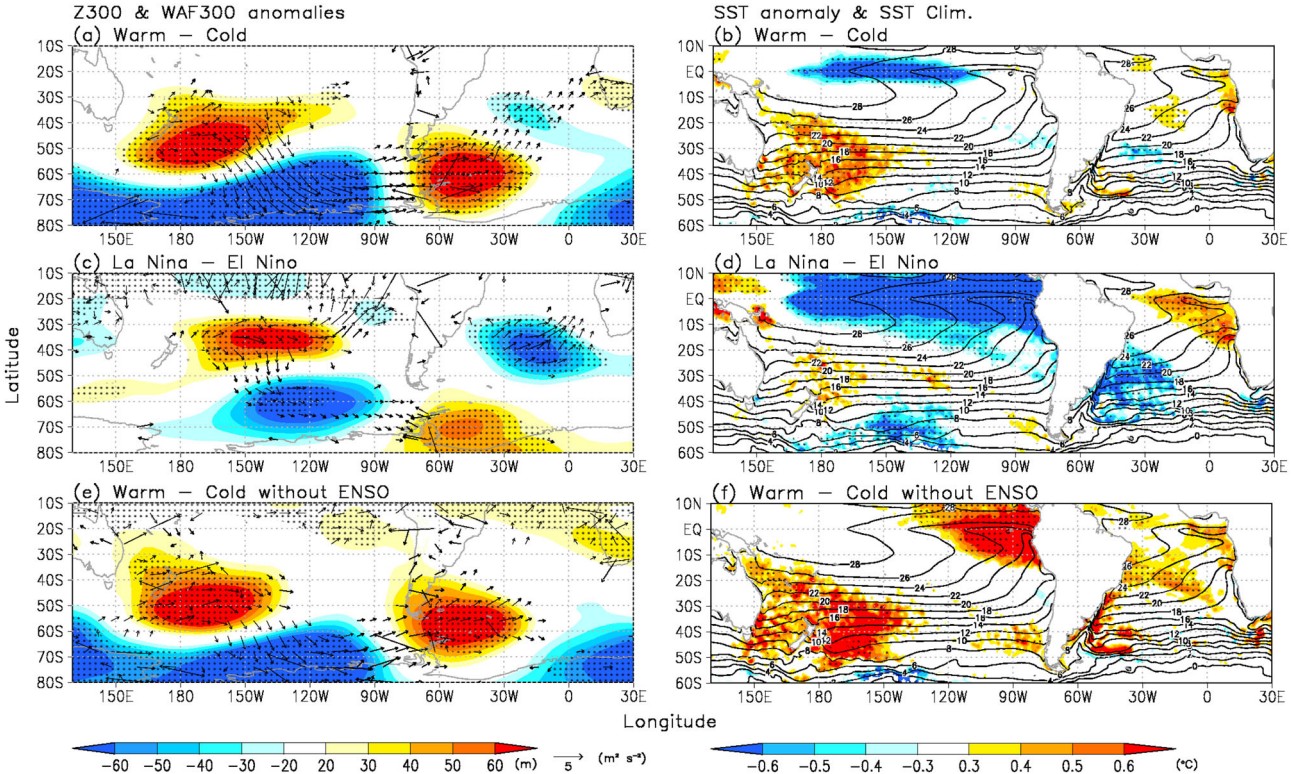

**Fig. 3 Atmospheric and ocean anomalies.** Difference maps for (**a**) geopotential height at 300 hPa (Z300) (m: shaded) with the horizontal component of wave-activity flux anomalies (m²/s²: vector) at 300 hPa (WAF300) from ref. [53] and (**b**) sea surface temperature (SST) differences between warm and cold winters. Black contours show climate values from 1979 to 2019 for SST. Dotted areas denote significant differences exceeding the 95% confidence level. **c, d, e, f** Same as **a, b**, but for between La Niña and El Niño winters (**c, d**) and between warm and cold winters without strong ENSO (**e, f**).

south of New Zealand to West Antarctic is clearly observed as in warm AP winters (Supplementary Fig. 7c). The SSTs show a warming anomaly over the Tasman Sea and a negative anomaly over the tropical Pacific region, meaning that both SST anomalies have impacts on these atmospheric circulation anomalies. In contrast, in negative IOD winters, there are no clear wave-like anomalies over the SH (Supplementary Fig. 7e) because atmospheric anomalies over the SH linked to the IOD are found during late winter and early spring[41].

In strong La Niña and positive SAM winters, the atmospheric circulation anomalies have relatively small amplitudes (Fig. 3a, c, and Supplementary Figs. 6a, b, d, and 7a). The SST anomalies in the Tasman Sea are absent in strong La Niña and positive SAM winters (Fig. 3d and Supplementary Fig. 7b). These results suggest that the SST warming over the Tasman Sea contributed to change in SH circulation.

**Atmospheric response to the forcing over the mid-latitudes in the western Pacific.** To address the atmospheric response to the SST anomalies over the Tasman Sea during warm and cold AP winters from a theoretical point of view, we conducted experiments using the AFES[62,63] (see "Methods" section). All experiments used atmospheric conditions of 1 June of each of the 41 years 1979–2019 as initial conditions and integrated for 3 winter months (June to August), meaning that each experiment has 41-ensemble members. All experiment results were based on 3-month averages over these 41 years. For the control experiment (CTL), climatological daily SST and sea ice cover data were used as boundary conditions. The SST anomalies over the globe between warm and cold AP winters superposed on the daily global climatology were used as forcing for a global experiment

(Globe) (Supplementary Fig. 8). Features of the Z300 response differences between Globe and CTL were a positive anomaly of Z300 east of Argentina and a negative anomaly over the Amundsen Sea, resulting in anomalous northerly warm advection in the lower troposphere over West Antarctica (Fig. 4a, Supplementary Fig. 9a). This anomalous pattern was similar to that observed, although the amplitudes of Z300 were smaller than those exhibited by the CFSR (Fig. 3a, Supplementary Fig. 6a).

To assess the impacts of the removal of certain tropical and Northern Hemisphere oceanic anomalies on atmospheric circulations in the SH, we ran further experiments that were forced by SST anomalies restricted to the tropical Pacific region (5°S–5°S, 170°–260°E) (ENSO experiment) and the Tasman Sea region (48–32°S, 160°–180°E) (TAS experiment) superposed on the daily global climatology (Supplementary Fig. 8 and see "Methods" section). We constructed difference maps in atmospheric circulations between the sensitivity experiments (Globe, ENSO, and TAS) and CTL.

The atmospheric response to only tropical Pacific cooling is similar to that shown in Fig. 4a (Fig. 4b, Supplementary Fig. 9b). However, in the TAS experiment, a wave-like anomaly pattern appeared from the subtropical Pacific to Antarctica in the Pacific sector, even without tropical SST forcing (Fig. 4c, Supplementary Fig. 9c). In particular, strong positive (negative) Z300 responses were observed over the east coast of Argentina (Amundsen Sea). These anomaly patterns, which induced AP warming, also originated from SST warming over the Tasman Sea. The atmospheric response to these anomalies included ridges off the southeast coast of Argentina and slightly downstream of the Drake Passage and a trough over the Amundsen Sea. The positive Z300 anomalies over the south of New Zealand and east coast of Argentina were induced by a poleward shift of cyclone tracks

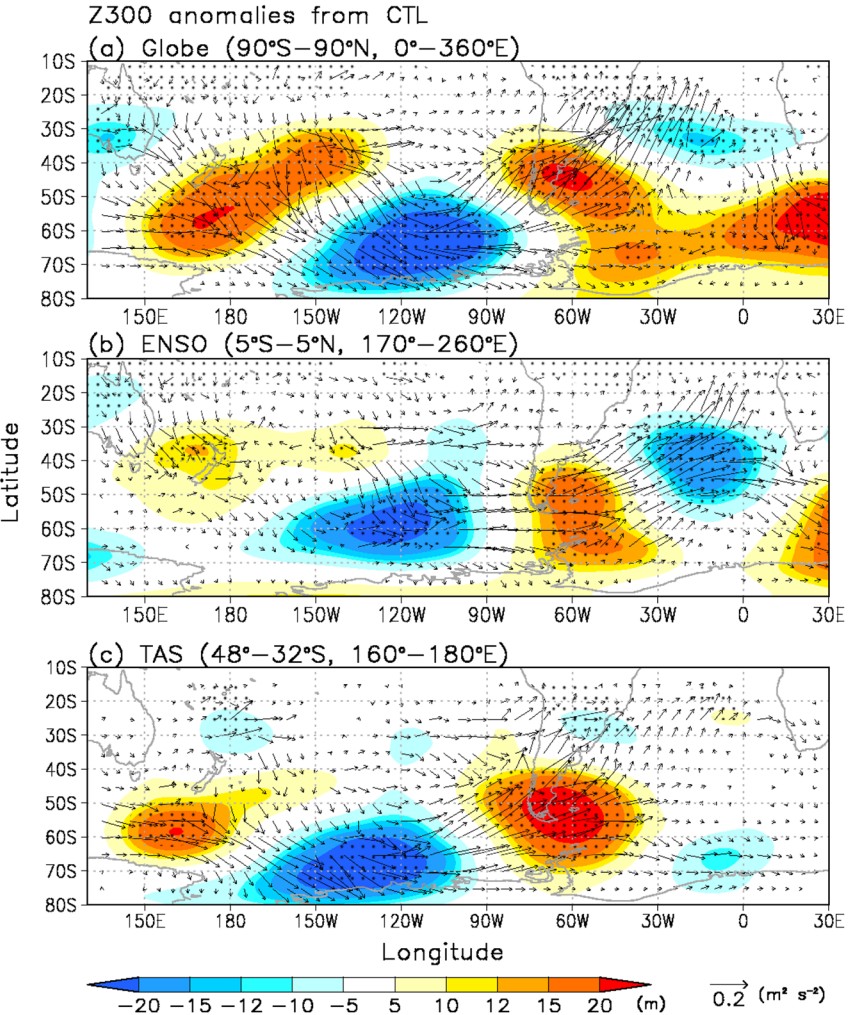

**Fig. 4 Simulated atmospheric responses to SST anomalies.** Geopotential height at 300 hPa (Z300) (m: shaded) with the horizontal component of wave-activity flux anomalies (m²/s²: vector) at 300 hPa (WAF300) from ref. [53] in (**a**) Globe–control experiment (CTL) difference, (**b**) ENSO–CTL difference and (**c**) TAS–CTL difference. For the CTL, climatological daily SST and sea ice cover data were used as lower boundary conditions. For each experiment (Globe, ENSO, and TAS experiment), the SST anomalies between warm and cold winters over each region (Globe: 90°S–90°N, 0–360°E, ENSO: 5°S–5°N, 170–260°E, TAS: 48–32°S, 160–180°E) superposed on the daily global climatology were used as forcing. Dotted areas denote significant differences exceeding the 95% confidence level.

related to the warm anomaly over the Tasman Sea, leading to northerly warm advection over the AP. To explore the statistical robustness of the results of the TAS experiment, we increase the numbers of ensemble members for CTL and TAS using initial conditions for two additional days (2 and 3 June in each of the 41 years), then integrated for 3 winter months as before. We diagnose a very similar response with this 123-member ensemble (Supplementary Fig. 10), indicating that our sample size of 41 is more than sufficient for establishing statistical significance.) On the basis of these results, the tropical SST cooling associated with ENSO has an important role in AP warming. However, the Tasman Sea SST anomalies alone produce warm winters in the AP.

## Discussion

In summary, we have shown that warm winter episodes in the Tasman Sea influence warm temperature anomalies over key regions of West Antarctica, including the AP, through a poleward shift of South Pacific cyclone tracks. The AFES model results present new insights into remote atmospheric responses to SH mid-latitude perturbations and the role that these may play in

influencing this sensitive region of West Antarctica. This broad concept is consistent, though in another context, with an analysis that showed that SH synoptic perturbations in the mid-latitudes tend to lead to variations in the Hadley Cell width[64].

The dramatic increases in marine heatwaves in the Tasman Sea have a potential impact on changes in atmospheric circulation over the SH in other seasons. To explore this for the three other seasons, we calculated difference maps of the atmospheric circulation and SST between warm and cold AP years for which the magnitudes of the temperature anomaly values, as before, exceeded one half standard deviation at six AP stations (Supplementary Figs. 11, 12). In warm spring (September to November) and autumn (March to May) years, the atmospheric circulation anomalies that induced warming over the AP resemble those in warm winters (Fig. 3a and Supplementary Fig. 12a, d). In these spring years, Tasman Sea warming and central Pacific cooling SST patterns were similar to those in warm winter years (Fig. 3b and Supplementary Fig. 12c). In contrast, in warm autumn years, there was significant warming over the Tasman Sea without tropical cooling (Supplementary Fig. 12f). On the basis of these results, although the SH atmospheric circulation anomalies were related to both SST anomalies in spring

years, the AP warming was induced by SST warming over the Tasman Sea even without anomalous tropical SST cooling in warm autumn years. In warm AP summers (December to February), there were small differences in surface temperature at the six AP stations between warm and cold summers (Supplementary Fig. 11c). Therefore, the atmospheric responses to the tropical Pacific SST cooling were not clearly observed over the SH as reported in previous studies (Supplementary Fig. 12g–i). Our study has particular relevance because the Tasman Sea has been identified as one that impacts the AP climate, except for in summer.

## Methods

**Observation data**. The Antarctic Climate Data, which includes temperature, surface pressure, and winds observed at manned and automatic weather stations, are available from the SCAR READER project (https://legacy.bas.ac.uk/met/READER)[65]. In this study, we selected six stations (Bellingshausen, O'Higgins, Esperanza, Marambio, Vernadsky, and Rothera) on the AP. Monthly averaged surface temperature data from these six AP stations were used in Fig. 1a. From this time series, we selected anomalous warm and cold winters (June to August) for which the magnitudes of the temperature anomaly values exceeded one half of a standard deviation (warm winters: 1983–1985, 1989, 1998, 2000, 2004, 2008, 2010, 2014, 2016, and 2018–2019; cold winters: 1980, 1986–1987, 1991–1992, 1994–1995, 2002, 2007, 2009, 2011 and 2015). In addition, for the other three seasons (spring [September to November], summer [December to February], and autumn [March and May]), we similarly selected warm and cold years (Supplementary Fig. 11).

**Reanalysis data**. We used 6-h data of the Climate Forecast System Reanalysis[49] (CFSR) from January 1979 to March 2011 and CFS Version 2[50] from April 2011. The CFSR and CFS data on a 0.5° horizontal grid is provided by the National Centers for Environmental Prediction (NCEP). We used meteorological (SLP, air temperature, wind speed, geopotential height, and precipitation) and oceanographic (sea ice concentration, sea surface temperature, and ice drifting speed) parameters. The wave activity flux representing stationary Rossby wave propagation was based on ref. [53]. In this study, the significance test was a standard two-tailed t-test with degrees of freedom based on the number of years in the figures and supplementary figures.

It is important to establish that our results do not depend on this specific choice of reanalysis set. Accordingly, we also made use of the European Centre for Medium-Range Weather Forecasts (ECMWF) ERA5 compilation[66] with a horizontal resolution of 0.5° × 0.5°. We constructed difference maps of T700, SLP, Z300, and wave activity fluxes at 300 hPa (WAF300) with the ERA5 data in the same form as in Fig. 2 (Supplementary Fig. 13). To reduce uncertainty in reanalysis data, the means of the 10 ensemble members of ERA5 were used. The plots differed little from those constructed with the NCEP product.

**Cyclone tracking data**. We used cyclone tracking data derived from an algorithm developed at the University of Melbourne[67]. This algorithm identifies the cyclone positions at 6-h intervals based on the Laplacian of pressure at each grid. Cyclones lasting less than 24 h were removed from the analysis. To investigate cyclone density, we calculated the densities of cyclones in a circular grid[68] with circular cells of a 5° latitude radius (equivalent to approximately 308,025 km²) (units: count/S) for each year.

**NINO3.4 index**. Monthly averaged SST anomalies were calculated for the NINO3.4 region (5°S–5°N, 190°–240°E). For our June to August period (Supplementary Fig. 5a), analogous to above, we identified strong La Niña and El Niño winters for which the NINO3.4 index values exceeded one half standard deviation (strong La Niña winters: 1981, 1984–1985, 1988–1989, 1995–1996, 1998–2000, and 2010; strong El Niño winters: 1982, 1987, 1991, 1997, 2002, 2004, 2009, 2012, and 2015). Using the CFSR data, we constructed difference maps of T700 with SLP, Z300 with WAF300, and SST with climatology SST values between La Niña and El Niño winters, as shown in Fig. 3a, b and Supplementary Fig. 6a (Fig. 3c, d and Supplementary Fig. 6b).

**Southern Annular Mode (SAM) index**. Observation-based Southern Hemisphere Annular Mode Index data[69] were used as the SAM index (available from https://legacy.bas.ac.uk/met/gjma/sam.html). From this anomaly time series for June to August (Supplementary Fig. 5c), we extracted strong positive and negative SAM winters, defined as those for which the magnitude of the SAM index values exceeded one half standard deviation (strong SAM+ winters: 1979, 1982, 1989, 1993, 1997, 1998, 2004, 2008, 2010, 2012, and 2015–2016; strong SAM– winters: 1981, 1984, 1988, 1990–1992, 1994–1996, 2000, 2007, 2009, 2011, 2013, and 2018–2019). Using the CFSR data, we constructed difference maps of T700 with

SLP, Z300 with WAF300, and SST with climatology SST values between strong SAM+ and SAM– winters, as shown in Fig. 3a, b and Supplementary Fig. 6a (Supplementary Figs. 6d and 7a, b).

**Indian Ocean dipole index**. The Indian Ocean Dipole (IOD) index[70] is defined as the SST anomaly difference between the tropical western Indian Ocean (10°S–10°N, 50°–70°E) and tropical southeast Indian Ocean (10°S–0°, 90°–110°E) (Supplementary Fig. 5e). From this anomaly time series for June to August, we selected strong positive and negative IOD winters for which the IOD index values exceeded one standard deviation (strong IOD+ winters: 1982–1983, 1994, 1997, 2003, 2007–08, 2012, 2015, and 2017–2019; strong IOD– winters: 1980–1981, 1984–1986, 1989, 1992, 1995–1996, 1998, 2001–2002, 2004–2005, 2009, 2013–2014, and 2016). Using the CFSR data, we constructed difference maps of T700 with SLP, Z300 with WAF300, and SST with climatology SST values between strong IOD– and IOD + winters, as shown in Fig. 3a, b and Supplementary Fig. 6a (Supplementary Figs. 6f, 7e, f).

**Removal of the impacts of ENSO and SAM**. To remove the impact of ENSO in warm and cold winters, we composited the atmospheric and ocean fields for warm and cold AP winters without strong ENSO winters (e.g., strong La Niña [1984–1985, 1989, 1998, 2000, and 2010] and strong El Niño [1987, 1991, 2002, 2009, and 2015] winters) (Supplementary Fig. 5b). We then formed the difference maps of these (Fig. 3e, f, and Supplementary Fig. 6c). In a similar fashion, we removed the impact of the SAM on warm and cold AP winters by assembling different maps of the atmosphere and ocean between warm and cold AP winters without a strong SAM (e.g., SAM+ [1989, 1998, 2004, 2008, 2010, and 2016] and SAM– (1991–1992, 1994–1995, 2007, 2009, 2011) winters) (Supplementary Figs. 5d, 6e, 7c, d).

**The atmospheric general circulation model For Earth Simulator (AFES)**. To conduct sensitivity experiments, we used the Atmospheric general circulation model For Earth Simulator (AFES)[62,63]. The AFES with a horizontal resolution of T119 (triangular spectral truncation with a truncation wavenumber of 119, ~1° × 1°) and 48 vertical levels reproduced the geopotential height and temperature structures of large-scale circulation in the troposphere and lower stratosphere, as well as other reanalysis products[71–73]. In this study, we ran all experiments initialized with atmospheric conditions of 1 June of each year (1979–2019) and integrated for 3 winter months (June to August). For the control experiment (CTL), climatological daily SST and sea ice cover data were used as lower boundary conditions. Supplementary Fig. 8 shows the difference in SSTs between warm and cold winters with climatological SST and sea ice cover in the winter months. To calculate the atmospheric response to SST anomalies over the entire globe (90°S–90°N, 0–360°E), the tropical region (5°S–5°N, 170–260°E; orange line in Supplementary Fig. 8) and the Tasman Sea area (48–32°S, 160–180°E; purple line in Supplementary Fig. 8), the SST anomalies between warm and cold winters over each region superposed on the daily global climatology were used as forcing for each experiment. For all experiments, results were based on 3-month averages from 1979 to 2019. The ERA5 data were used as atmospheric initial conditions and boundary conditions for the AFES.

## Data availability

The surface temperature datasets were obtained from the SCAR READER project (https://legacy.bas.ac.uk/met/READER). We used the observation-based Southern Hemisphere Annular Mode Index (https://legacy.bas.ac.uk/met/gjma/sam.html). To composite atmospheric and oceanic maps, we used the CFSR (https://rda.ucar.edu/) and ERA5 (available from https://www.ecmwf.int/) reanalyses provided by the NCEP and ECMWF. AFES experiment data are available from the corresponding author on request.

## Code availability

All codes used to analyze and plot the data are available from the corresponding author on request.

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

## Acknowledgements

This work was supported by a JSPS Overseas Research Fellowship, JSPS KAKENHI (20H04963, 19K14802, 18H05053). Parts of this research were made possible by funding from the Australian Research Council (Grant DP16010997). The AFES integration was performed on the Earth Simulator with the support of JAMSTEC. We thank Sara J. Mason for correcting a draft of this manuscript.

## Author contributions

K.S. and J.I. designed the research. K.S., J.I., and I.S. wrote the paper. I.R. made cyclone track data sets. K.S. conducted the experiments using the AFES. All authors commented on the manuscript.

## Competing interests

The authors declare no competing interests.
