## [Peer Review File · Nature Communications]

Reviewer first round comments:

Reviewers' comments:

Reviewer #1 (Remarks to the Author):

This research continues recent attempts to find the causes of June-July Antarctic Peninsula climate variability. I enjoyed reading this manuscript and think that it is a good piece of work that is going in the right direction to better understand SH climate variability. Although I am inclined to believe that tropical Pacific SST is more important than SST in other tropical and extratropical basins to influence the SH circulation, I am convinced by some evidence here that midlatitude Pacific SST may also play a role in the connection. However, there are a few things that could be done to strengthen the paper, and so I recommend minor revision.

What I learned from classical climate dynamics tells me that tropical SST is always more efficient to drive extratropical circulation than extratropical SST. I think this concept may also hold well for this case. However, the LBM results are pretty convincing in the paper. So I think one argument to reconcile these is that the warm SST anomaly close to the Tasman Sea could disturb the extratropical jet along 30S, which as we know acts as an effective wave source to emanate wave train propagating downstream to the West Antarctic (please see Ding et al. 2012, <https://journals.ametsoc.org/doi/full/10.1175/JCLI-D-11-00523.1>). I think the wave activity analysis presented in this paper (Fig.2b) also supports this idea.

I don't understand why the authors collectively use 13 different basic states in the LBM rather than one mean state averaged over the entire period (state 13 in table 1). I think the authors' approach should be justified. The LBM is a linearized version of the primitive equations. Thus, it makes more sense to me that the model should use one mean state that is averaged over an enough long period.

Reviewer #2 (Remarks to the Author):

Review on "Antarctic Peninsula warm winters influenced by Tasman Sea temperatures" by Kazutoshi Sato, Jun Inoue, Ian Simmonds, and Irina Rudeva.

This study investigates how warm SSTs in the Tasman Sea influence the June-July temperatures at Bellingshausen station located north of the Antarctic Peninsula. The relationship seems plausible, however, there is a significant lack of evidence demonstrating a robust physical connection. In addition, many important details are missing from the author's interpretation of their observational analysis, and it is not appropriately placed in the context of prior findings. For these reasons, and several others detailed below, I do not feel this study is suitable for publication in Nature Communications.

Major comments

Fig. 2b and Lines 99-102

The observed circulation clearly reflects a +SAM/zonal wave 3 pattern with an embedded La Niña/PSA. The La Niña/PSA pattern is further demonstrated in Fig. 2b by the strong poleward wave activity coming from the central tropical Pacific east of New Zealand and the significant La Niña SST anomaly pattern in Supplementary Fig. 1. This is consistent with previous studies (Clem and Fogt 2013, Fogt and Bromwich 2006) that have shown in-phase La Niña/+SAM events amplify the ABSL and lead to stronger widespread temperature anomalies on the Antarctic Peninsula. Furthermore, many studies have linked tropical Pacific climate variability to the Peninsula climate (Clem and Fogt 2013, Ding and Steig 2013, Clem and Fogt 2015, Clem et al. 2016, Turner et al. 2016), consistent with the wave activity pattern in Fig. 2b. Despite this, Fig. 2b is summarized as: "The propagation of wave activity flux shows a wave train originating from the south of the New Zealand region. This indicates that these wave-like anomaly patterns are associated with the

midlatitudes variability, particularly changes in SST in the Tasman Sea." One cannot simply ignore the wave activity coming out of the central tropical Pacific and throw out all prior knowledge of the strong relationship between Peninsula climate and tropical variability. Moreover, this study never goes on to even show a wave train originating south of New Zealand as is claimed here. And the claim that the pattern shown in Fig. 2b is linked to changes in Tasman Sea SSTs is completely unjustified here. Instead, in my opinion, Fig. 2b shows equatorward wave fluxes/poleward momentum fluxes over the Tasman Sea/New Zealand region originating at high latitudes south of Australia, i.e. associated with the upstream storm track. This is consistent with prior studies of L'Heureux and Thompson 2006 and Fogt et al. 2011 that +SAM and La Niña increase poleward momentum fluxes in turn strengthening the polar jet, as is later shown in Fig. 4b.

Lines 125-126: why would this be "expected" to influence the Peninsula?

Your interpretation is unclear and not totally accurate, e.g. lines 134-137, one cannot simply say there is a poleward shift of the westerlies without quantifying it...in fact, in some places like south of Australia and over the Pacific in Fig. 4b, I see an equatorward shift. Moreover, it is not clear what "change in the atmospheric heating in the downstream regions" means.

Your linear baroclinic model experiments do not convince me that the Tasman Sea is important to warming at Bellingshausen. In Fig. 5a I don't see any heating in the Tasman Sea used to force the model (perhaps your experimental design is not clear?). For the Tasman Sea-only experiment, the circulation pattern is weak, and in my opinion it does not resemble the observed circulation pattern nor a circulation pattern in general that would warm Bellingshausen. For example, the anticyclone to the northwest of the Peninsula (rather than off the east coast of Argentina) would likely cool the Peninsula rather than warm it, especially with the small deepened ABSL located very far west of the Peninsula nearly in the Ross Sea. Furthermore, the fact that there is no strong circulation response in your tropics-only experiment, despite the observational results and prior studies showing a strong influence from the tropics, your experimental setup is likely missing important non-linear processes going on in the real circulation pattern...one cannot simply say the tropics are not important. For these reasons I am not convinced that the Tasman Sea plays an important, especially not a more important, role than the tropics or SAM or zonal wave 3, etc. It is plausible to me that warming in the Tasman Sea could influence the Peninsula, but again this needs more concrete evidence.

Minor points

Given there are nearly twice as many supplementary figures-many of which are important in my opinion-as main figures, perhaps this work would be more suitable in a longer journal that would allow for a more detailed discussion and interpretation of the role of the Tasman Sea compared to the SAM/tropics/zonal wave 3, which are also important.

Your experimental design is not very clear especially with regards to how you are isolating heating in the Tasman Sea.

How did you account for statistical significance in your model results?

A bit pedantic, but technically Bellingshausen is not on the Antarctic Peninsula, making your title misleading. Do you see similar results for stations located on the Antarctic Peninsula?

What is the primary local mechanism leading to warming at Bellingshausen...is it sea ice anomalies (i.e., local air-sea heat fluxes), thermal advection, something else?

What makes June-July so unique that the entire study focuses on only two months? Yes it is true that the mean background circulation is more favorable for Rossby wave propagation out of the tropics during winter, but so too is autumn (Ding and Steig 2013) and spring (Schneider et al. 2012; Clem and Fogt 2015). It isn't clear to me why only two months are presented here.

I would like to see a time series of Tasman Sea SST alongside Bellingshausen temperature (and other Peninsula station temperature/climate parameter). Has warming of the Tasman Sea overlapped with warming at Bellingshausen? If the relationship is indeed linear, I would expect to

see a relationship between warming at Bellingshausen and warming SSTs in the Tasman Sea.

References

- Clem, K. R., J. A. Renwick, J. McGregor, and R. L. Fogt (2016), The relative influence of ENSO and SAM on Antarctic Peninsula climate, *J. Geophys. Res. Atmos.*, 121, 9324–9341, doi:10.1002/2016JD025305.
- Clem, K. R., and R. L. Fogt, 2013: Varying roles of ENSO and SAM on Antarctic Peninsula climate. *J. Geophys. Res. Atmos.*, doi:10.1002/jgrd.50860
- Clem, K. R., and R. L. Fogt, 2015: South Pacific circulation changes and their connection to the tropics and regional Antarctic warming in austral spring, 1979–2012. *J. Geophys. Res. Atmos.*, 120, 2773–2792, doi:10.1002/2014JD022940.
- Fogt R. L., and D. H. Bromwich, 2006: Decadal variability of the ENSO teleconnection to the high latitude South Pacific governed by coupling with the Southern Annular Mode. *J. Climate*, 19, 979–997.
- Fogt, R. L., D. H. Bromwich, and K. M. Hines, 2011: Understanding the SAM influence on the South Pacific ENSO teleconnection. *Clim. Dyn.*, 36, 1555–1576.
- Ding, Q., and E. J. Steig, 2013: Temperature change on the Antarctic Peninsula linked to the tropical Pacific. *J. Climate*, 26, 7570–7585, doi:10.1175/JCLI-D-12-00729.1.
- Schneider, D. P., C. Deser, and Y. Okumura, 2012a: An assessment and interpretation of the observed warming of West Antarctica in the austral spring. *Clim. Dyn.*, 38, 323–347.
- L’Heureux, M. L., and D. W. J. Thompson, 2006: Observed relationships between the El-Niño / Southern Oscillation and the extratropical zonal-mean circulation. *J. Climate*, 19, 276–287.
- Turner, J., Lu, H., White, I., King, J. C., Phillips, T., Hosking, J. S., et al. (2016). Absence of 21st century warming on Antarctic Peninsula consistent with natural variability. *Nature*, 535(7612), 411–415. <https://doi.org/10.1038/nature18645>

Reviewer #3 (Remarks to the Author):

This is a generally well written paper and is definitely of publication quality but I don't see anything sufficiently new or substantial here for it to be worthy of a place in a Nature publication. A lot of the analysis of atmospheric changes associated with warm Antarctic Peninsula winters follows a very similar methodology to a paper written more than 20 years ago (Marshall & King 1998 in GRL). While the specific influence of Tasman Sea temperatures in contributing to these changes and the associated model experiments are indeed new, I would have liked to see a greater discussion of how this finding related to other influences on Antarctic climate (other than ENSO and the SAM). For example, the Indian Ocean Dipole is also associated with changing SSTs in the Tasman Sea region and changing SLP in the ABS region of the Southern Ocean (e.g. Nuncio & Yuan 2015 doi: 10.1175/JCLI-D-14-00390.1, Na et al. 2006, doi: 10.1007/BF02842810) but is not mentioned.

Minor points

Line 14. Present tense is incorrect; e.g. Turner et al. (2016) (ref. # 11) show large warming ended at end of 20th Century.

Line 29. Similar to above, the use of 'since' suggests the dramatic temperature changes are still taking place. Also no explanation of 1957 (IGY?): e.g. there are local temperature records going back further, together with proxy temperatures from ice cores.

Line 30. I think refs. # 4 and 5 should be switched: e.g. ref. # 4 describes warming from radiosondes, not # 5.

Line 33. As currently written, this sentence contradicts the first sentence of this section (e.g. see my previous comments above).

Line 38. Marshall & Thompson (2016) doi:10.1002/2015JD024665 is an appropriate reference to add here given their Figure 2 shows heat flux anomalies associated with the SAM and other teleconnection patterns.

Line 38. What do the authors mean by 'strength' of the ABSL (now usually called simply ASL in most of the literature)? E.g. the absolute value of the pressure or its value relative to the surrounding region: see discussion in Hosking et al. (ref. # 15).

Line 53. Again see the difference between the PSA1 and PSA2 patterns in Fig. 2 of Marshall & Thompson (2016) as a good example of this.

Line 68. Given that the Antarctic Peninsula is one of the few places on the continent when there are actually a reasonable number of stations, it seems strange that the authors only use data from Bellingshausen. For example, they could use the set of six stations that Turner et al. (2016) do, with stations on both the west and east sides of the Peninsula, to provide a proper 'regional' set of temperature anomalies. All these stations are coastal low elevation sites. As an example, compare Fig. 1a to Fig. 1 of Marshall & King (1998) in GRL, who used Vernadsky station in an identical way, and the years with SAT more than 1 standard deviation from the mean are not the same (I appreciate that one is JJA and the other is JJ).

Line 74. Might be worth saying that CFSR has been validated in the Antarctic (e.g. Bracegirdle & Marshall (2012) doi: 10.1175/JCLI-D-11-00685.1) and found to do a reasonable job.

Line 134. To me Fig. 4b looks like changes in the split jet (see Bals-Elsholz et al. 2001 in J Clim.) e.g. a strengthening of the polar front jet and a weakening of the subtropical jet in warm winters. A similar result was shown by Marshall & King (1998). Why is the impact on the split jet and its effect on Peninsula SAT not mentioned?

Fig. 5. (b) and (c) should be swapped round to the usual layout of a 2 x 2 figure.

We would like to thank you for providing constructive comments regarding our manuscript. In revised form, we believe the modifications will be satisfactory and suitable for publication. The point-by-point responses are made below.

Reviewer #1 (Remarks to the Author):

This research continues recent attempts to find the causes of June-July Antarctic Peninsula climate variability. I enjoyed reading this manuscript and think that it is a good piece of work that is going in the right direction to better understand SH climate variability. Although I am inclined to believe that tropical Pacific SST is more important than SST in other tropical and extratropical basins to influence the SH circulation, I am convinced by some evidence here that midlatitude Pacific SST may also play a role in the connection. However, there are a few things that could be done to strengthen the paper, and so I recommend minor revision.

What I learned from classical climate dynamics tells me that tropical SST is always more efficient to drive extratropical circulation than extratropical SST. I think this concept may also hold well for this case. However, the LBM results are pretty convincing in the paper. So I think one argument to reconcile these is that the warm SST anomaly close to the Tasman Sea could disturb the extratropical jet along 30S, which as we know acts as an effective wave source to emanate wave train propagating downstream to the West Antarctic (please see Ding et al. 2012, <https://journals.ametsoc.org/doi/full/10.1175/JCLI-D-11-00523.1>). I think the wave activity analysis presented in this paper (Fig.2b) also supports this idea.

Thank you very much for pointing out the physics of the link between Tasman Sea SST and our observed wave train. In warm winters, wave activity flux anomalies appear to originate from subtropical region to Antarctica (Figure A1b). The negative SST anomaly is seen over the tropical region (Figure A1d), meaning that the tropical cooling anomaly influences on atmospheric anomalies over the Southern Hemisphere. However, the difference maps of geopotential height and wave activity flux at 300 hPa between strong La Niñas (1984, 1985, 1998, 1999 and 2010) and El Niños (1987, 97, 2002, 2009 and 2015) winters show the clear wave-like anomalies from tropical Pacific region to West Antarctica in strong La Niña winters (Figure A1f). In addition, the amplitudes and areas of negative differences in tropical Pacific SST between strong La Niña and El Niño winters are larger than those between warm and cold winters (Figure A1h). These results mean that the impact of tropical heating in warm AP winters would be weaker than that in La Niña winters. To remove the effects of the ENSO in warm and cold winters, we assembled composite maps of the atmospheric and ocean difference fields between warm and cold AP winters without strong ENSO winters (e.g. strong La Niña [1998, 2010] and El Niño [1987] winters) (Figures A1q-t, A2d). In warm winters without La Niña winters, the Antarctica Peninsula (AP) air temperature and Tasman Sea SST anomalies are positive, although negative SST anomalies are not seen over the tropical Pacific region (Figure A1t). In addition, there are no strong wave activity fluxes from tropical Pacific region to midlatitude (Figure A1r). In contrast, Z300 anomalies and wave activity fluxes remain strong over the downstream regions of Tasman Sea (e.g. Pacific sector of Southern Ocean), leading to intensification of polar jet over the south of New Zealand. These results indicated that tropical heating has an important role in AP warming. However, when tropical heating is weak, the Tasman Sea warming would influence the AP warming and the polar jet intensification over the south of New Zealand.

We added the sentences below (pages 11-12, lines 218-242):

“Previous investigations have shown that strengthening of the ABSL can be induced by the La Niña, positive SAM and negative IOD²²⁻⁴⁰. In warm AP winters, the wave activity flux anomalies appear to originate from subtropical Pacific to Antarctica, suggesting that the central Pacific cooling influences atmospheric circulation over the Southern Hemisphere as reported by previous studies. In the La Niña winters, the amplitudes and areas of tropical Pacific SST cooling are larger than those in warm winters (Methods, Supplementary Fig. 1h). These strong cooling anomalies contribute to the clear wave-like anomalies from the tropical Pacific region to West Antarctica (Supplementary Fig. 1e-h). In addition, in warm AP winters without strong El Niño, there are no negative tropical SST anomaly and wave activity flux from Pacific subtropical to West Antarctic (Methods, Supplementary Fig. 1q-t). Therefore, the amplitude of

atmospheric response to tropical Pacific cooling during warm AP winters would be weaker than that in strong La Niña winters, even without nonlinear process in the LBM. From these results, the tropical heating has an important role in Antarctic Peninsula warming. However, when tropical heating is weak, the Antarctic Peninsula warming is influenced by the Tasman Sea warming. In positive (stronger) SAM winters, the atmospheric circulation anomalies resemble those in warm winters over the Antarctic Peninsula (Supplementary Fig. 1i-l). The northeast Antarctic Peninsula warming, which has significant positive correlation with SAM³³, is seen clearly in strong positive SAM winters. However, in warm AP winters without strong SAM, the wave activity flux remains strong from south of New Zealand to west Antarctic (Supplementary Fig. 1u-x). In contrast, in negative IOD winters, there are no clear wave-like anomalies over the Southern Hemisphere (Supplementary Fig. 1m-p), because atmospheric anomalies over the Southern Hemisphere linked to the IOD are found during late winter and early spring^{40,41}.

And added the sentence to Methods (page 18, lines 355-362):

“Removal of the impacts of ENSO and SAM. To remove the impact of ENSO in warm and cold winters, we composited the atmospheric and ocean fields for warm and for cold AP winters without strong ENSO winters (e.g. strong La Niña [1998, 2010] and El Niño [1987] winters) (Supplementary Fig. 10d). We then formed the difference maps of these (Supplementary Fig. 1q-t). In a similar fashion we removed the impact of SAM on warm and cold AP winters by assembling the difference maps of atmosphere and ocean between warm and cold AP winters without strong SAM (e.g., SAM+ [1989, 2004, 2010, 2016] and SAM- (1992, 1995, 2007) winters) (Supplementary Figs. 1u-x, 10e).”

Figure A1: Difference maps for (a) air temperature at 700 hPa ($^{\circ}\text{C}$: shaded) and sea level pressure (hPa: contour), (b) geopotential height at 300 hPa (m: shaded) with the horizontal component of wave-activity flux anomalies (m^2/s^2 : vector) at 300 hPa, (c) U-wind speed (m/s: shaded) at 300 hPa (U300) and (d) sea surface temperature differences between warm and cold winters. Black contours show climate values from 1979 to 2018 for (c) U300 and (d) SST. Dotted areas denote significant differences exceeding the 95% confidence level. (e-h) (i-l) (m-p) (q-t) (u-x) Same as (a-d), but for between La Niña and El Niño winters (e-h), between positive SAM and negative SAM winters (i-l), between negative IOD and positive IOD winters (m-p), between warm and cold winters without strong ENSO winters (q-t), between warm and cold winters without strong ENSO winters and (u-x) between warm and cold winters without strong SAM winters.

Figure A2: Time series of (a) the NINO3.4 index, (b) the SAM index, (c) IOD index during June and July. Red and blue dots indicated strong El Niño, positive SAM, positive IOD winters and strong La Niña, negative SAM, negative IOD winters. (d) Time series of averaged surface air temperature anomalies ($^{\circ}\text{C}$, deviation from climatology for 1979–2018) during June and July at the Russian Bellingshausen station. Red and blue dots indicate the warm and cold winters for which the magnitude of the temperature anomaly values exceeded one standard deviation without strong El Niño and La Niña winters. (e) Same as (d), but for without positive and negative winters. Dashed lines show one standard deviation ($\pm 1\sigma$).

I don't understand why the authors collectively use 13 different basic states in the LBM rather than one mean state averaged over the entire period (state 13 in table 1). I think the authors' approach should be justified. The LBM is a linearized version of the primitive equations. Thus, it makes more sense to me that the model should use one mean state that is averaged over an enough long period.

As implied by reviewer, in most studies one mean state (in our case the average from 1979 to 2018) is used as the background flow for model studies. A problem with using the mean flow field is that it is not representative of any particular period (e.g. 1980s, 1990s, 2000s), and hence it is argued that our approach adds much more insight. Therefore, we conducted our experiment for an ensemble of background states in part to provide a measure of the robustness of the results to these equally-feasible background flows. In particular it allows us to test the statistical significance of the differences that the model simulates.

Reviewer #2 (Remarks to the Author):

Review on “Antarctic Peninsula warm winters influenced by Tasman Sea temperatures” by Kazutoshi Sato, Jun Inoue, Ian Simmonds, and Irina Rudeva.

This study investigates how warm SSTs in the Tasman Sea influence the June-July temperatures at Bellingshausen station located north of the Antarctic Peninsula. The relationship seems plausible, however, there is a significant lack of evidence demonstrating a robust physical connection. In addition, many important details are missing from the author's interpretation of their observational analysis, and it is not appropriately placed in the context of prior findings. For these reasons, and several others detailed below, I do not feel this study is suitable for publication in Nature Communications.

Major comments

Fig. 2b and Lines 99-102

The observed circulation clearly reflects a +SAM/zonal wave 3 pattern with an embedded La Niña/PSA. The La Niña/PSA pattern is further demonstrated in Fig. 2b by the strong poleward wave activity coming from the central tropical Pacific east of New Zealand and the significant La Niña SST anomaly pattern in Supplementary Fig. 1. This is consistent with previous studies (Clem and Fogt 2013, Fogt and Bromwich 2006) that have shown in-phase La Niña/+SAM events amplify the ABSL and lead to stronger widespread temperature anomalies on the Antarctic Peninsula. Furthermore, many studies have linked tropical Pacific climate variability to the Peninsula climate (Clem and Fogt 2013, Ding and Steig 2013, Clem and Fogt 2015, Clem et al. 2016, Turner et al. 2016), consistent with the wave activity pattern in Fig. 2b. Despite this, Fig. 2b is summarized as: "The propagation of wave activity flux shows a wave train originating from the south of the New Zealand region. This indicates that these wave-like anomaly patterns are associated with the midlatitudes variability, particularly changes in SST in the Tasman Sea." One cannot simply ignore the wave activity coming out of the central tropical Pacific and throw out all prior knowledge of the strong relationship between Peninsula climate and tropical variability. Moreover, this study never goes on to even show a wave train originating south of New Zealand as is claimed here. And the claim that the pattern shown in Fig. 2b is linked to changes in Tasman Sea SSTs is completely unjustified here. Instead, in my opinion, Fig. 2b shows equatorward wave fluxes/poleward momentum fluxes over the Tasman Sea/New Zealand region originating at high latitudes south of Australia, i.e. associated with the upstream storm track. This is consistent with prior studies of L'Heureux and Thompson 2006 and Fogt et al. 2011 that +SAM and La Niña increase poleward momentum fluxes in turn strengthening the polar jet, as is later shown in Fig. 4b.

Thank you very much for your perspectives. In warm winters, wave activity flux anomalies appear to originate from subtropical region to Antarctica (Figure A1b). The negative SST anomaly is seen over the tropical region (Figure A1d), meaning that the tropical cooling anomaly influences on atmospheric anomalies over the Southern Hemisphere. However, the difference maps of geopotential height and wave activity flux at 300 hPa between strong La Niña (1984, 1985, 1998, 1999 and 2010) and El Niño (1987, 97, 2002, 2009 and 2015) winters show the clear wave-like anomalies from the tropical Pacific region to West Antarctica in strong La Niña winters (Figure A1f). In addition, the amplitudes and areas of negative differences in tropical Pacific SST between strong La Niña and El Niño winters are larger than those between warm and cold winters (Figure A1h). These results indicate that the impact of tropical heating in warm AP winters would be weaker than that in La Niña winters. To remove the effects of the ENSO in warm and cold AP winters, we made the atmospheric and oceanic fields between warm and cold winters without strong ENSO winters (e.g. strong La Niña [1998, 2010] and El Niño [1987] winters) (Figures A1q-t, A2d). In warm winters without La Niña winters, the Antarctica Peninsula (AP) air temperature and Tasman Sea SST anomalies are positive, although negative SST anomalies are not seen over the tropical Pacific region (Figure A1t). In addition, there are no strong wave activity fluxes from tropical Pacific region to midlatitude (Figure A1r). In contrast, Z300 anomalies and wave activity fluxes remain strong over the downstream regions of Tasman Sea (e.g. Pacific sector of Southern Ocean), leading to intensification of polar jet over the south of New Zealand. These results indicated that tropical heating has an important role in

AP warming. However, when tropical heating is weak, the Tasman Sea warming would influence the AP warming and the polar jet intensification over the south of New Zealand.

Next, to remove the impact of the SAM on AP warming, we constructed difference maps of atmospheric and oceanic fields between warm and cold winters without strong positive (1989, 2004, 2010, 2016) and negative (1992, 1995, 2007) SAM winters from warm and cold winters (Figures A1u-x, A2e). Although wave activity flux anomaly related to tropical cooling anomaly appears from tropical region to Pacific Sector of Southern Ocean, there are activity flux anomalies from positive Z300 anomaly over the south part of New Zealand. In addition, the Tasman Sea has significant warm SST anomaly. These results mean that the AP warming is influenced by not only tropical heating anomaly but also Tasman Sea warming when the strong SAM effects are absent.

There are wave activity flux anomalies from Antarctica to south of New Zealand between positive and negative SAM winters. In contrast, in warm winters without strong SAM winters, the wave activity flux anomalies are unclear from Antarctica to south of New Zealand. Therefore, the SAM would have an impact on these wave activity anomalies. However, these anomalies are not clearly seen from Antarctica to south of New Zealand compared with over the Pacific sector of the Southern Ocean, meaning that these anomalies have minor role in atmospheric circulation anomalies.

As suggested by reviewer, in the Southern Ocean, the strong wave activity fluxes appear to originate from positive Z300 anomaly over the south part of New Zealand, not over the Tasman Sea. However, this positive Z300 anomaly is induced by the poleward shifts of cyclone tracks over the South parts of New Zealand and Australia. Therefore, the wave activity flux anomaly is associated with Tasman Sea warming.

We added the sentences about discussing another influences (pages 11-12, lines 218-242). For details, please see reply for comment #1 of reviewer #1.

Lines 125-126: why would this be "expected" to influence the Peninsula?

We revised the sentence below (page 8, lines 137-139):

“Therefore, a warming Tasman Sea has strengthened the SST gradient between high and mid latitudes (increased baroclinicity), and hence has influenced on cyclone tracks over the Southern Ocean.”

Your interpretation is unclear and not totally accurate, e.g. lines 134-137, one cannot simply say there is a poleward shift of the westerlies without quantifying it...in fact, in some places like south of Australia and over the Pacific in Fig. 4b, I see an equatorward shift. Moreover, it is not clear what "change in the atmospheric heating in the downstream regions" means.

Figure A3 shows the composited U-wind speed and geopotential height at 300 hPa (U300 and Z300) for (a) warm and (b) cold winters. Over the Pacific sector of the Southern Ocean, the positive U300 anomaly would come from the poleward shift of extratropical jet. To the south of New Zealand and Australia, the U300s in warm winters are stronger than those in cold winters. As clearly seen in the two parts of Fig. A3, the polar jet shifts poleward between 150 and 210°E.

We added the sentence below (page 8, lines 144-148) and figure:

“The zonal wind speed at 300hPa has strongest positive anomaly to the south of New Zealand (Fig. 4b), similar to split jet patterns with strong polar jets^{27,60}. However, in warm winters, the polar jet tends to shift poleward over the region south of Australia and New Zealand (Supplementary Figs. 4, 5b).”

And revised the sentences (lines 152-153)

“...over the Tasman Sea, causing change in the atmospheric heating in the downstream regions (i.e. Pacific sector of the Southern Ocean).”

Figure A3: Composited U-wind speed (m/s: shaded) and geopotential height (m: contour) at 300 hPa for (a) warm and (b) cold winters.

Your linear baroclinic model experiments do not convince me that the Tasman Sea is important to warming at Bellingshausen. In Fig. 5a I don't see any heating in the Tasman Sea used to force the model (perhaps your experimental design is not clear?). For the Tasman Sea-only experiment, the circulation pattern is weak, and in my opinion it does not resemble the observed circulation pattern nor a circulation pattern in general that would warm Bellingshausen. For example, the anticyclone to the northwest of the Peninsula (rather than off the east coast of Argentina) would likely cool the Peninsula rather than warm it, especially with the small deepened ABSL located very far west of the Peninsula nearly in the Ross Sea. Furthermore, the fact that there is no strong circulation response in your tropics-only experiment, despite the observational results and prior studies showing a strong influence from the tropics, your experimental setup is likely missing important non-linear processes going on in the real circulation pattern...one cannot simply say the tropics are not important. For these reasons I am not convinced that the Tasman Sea plays an important, especially not a more important, role than the tropics or SAM or zonal wave 3, etc. It is plausible to me that warming in the Tasman Sea could influence the Peninsula, but again this needs more concrete evidence.

The atmospheric response to the heating anomaly over the Tasman Sea area has a small direct impact on AP warming. According to the experiment in which heating anomaly is placed over the Amundsen Sea and Drake Passage, the cyclonic anomaly related to cooling anomaly over the Amundsen Sea and the anticyclonic anomaly related to heating anomaly over the Drake Passage have important roles in the AP warming. However, the cold advection, which induced by cyclone anomaly related to heating anomaly over the Tasman Sea area, causes the atmospheric cooling over the Amundsen Sea. The physical mechanisms can be explained as follows:

- 1) The poleward shift of upper-level wind (polar jet) over the south of Australia and New Zealand are induced by enhanced SST gradient related to Tasman Sea warming, leading to poleward shift of cyclone tracks over these regions.
- 2) The poleward shift of cyclone tracks causes negative Z300 anomaly coast of Antarctica and positive Z300 anomaly over the east of New Zealand. The changes in horizontal advection related to these atmospheric circulation anomaly causes atmospheric heating (Q1) anomalies over the Tasman Sea area.
- 3) Atmospheric responses to Q1 anomalies over the Tasman Sea area have cyclonic anomaly over the Amundsen Sea and anticyclonic anomaly across the Pacific at 45-55°S. Over south of New Zealand, warm advection related to anticyclonic anomaly over the Pacific sector of Southern Ocean induced atmospheric heating, strengthening further positive Z300 anomaly over the east of New Zealand. In contrast, this cyclonic anomaly leads to cold advection from Antarctica to the Amundsen Sea, causing strong atmospheric cooling over the Amundsen Sea.
- 4) The atmospheric cooling over the Amundsen Sea make strong cyclonic anomaly, which causes warm advection over the AP and cold advection over the Amundsen Sea. This warm advection contributes to the AP warming. This cold advection causes further atmospheric cooling over the Amundsen Sea.

Therefore, the AP warming would be triggered by the atmospheric responses to heating anomaly associated with Tasman Sea warming.

We added the sentence below (pages 10-11, lines 189-204):

“The positive Z300 anomaly over the east of New Zealand and negative Z300 anomaly over the south of New Zealand and Australia are induced by poleward shift of cyclone tracks related to Tasman Sea warming, leading to atmospheric heating anomalies over the Tasman Sea sector (Fig. 2b, Supplementary Fig. 7b). The atmospheric response to the heating anomaly over the Tasman Sea sector includes a strong ridge across the Pacific at 45-55°S and trough over the Amundsen Sea (Supplementary Fig. 7b). This ridge is reminiscent of that in the global-forcing experiment (Fig 5b), but is not seen in any of the other sector experiments shown in Supplementary Fig. 7. In contrast, the amplitude of trough is smaller than that in the Amundsen Sea-forcing experiment (Supplementary Fig. 7b,c). The strongest negative Z300 responses, which induced Antarctic Peninsula warming, came from Q1 cooling over the Amundsen sector (Supplementary Fig. 7c). However, relatively weak trough associated with Tasman Sea forcing causes southerly cold advection from

Antarctica, resulting in cooling anomaly over the Amundsen Sea (Supplementary Fig. 7b). These experiments indicate that the Antarctic Peninsula warming is triggered by the Tasman Sea warming.”

The non-linear process is not treated in the LBM. Therefore, the amplitudes of atmospheric response to heating anomalies over the entire globe are smaller than observed one, in particular the positive Z300 anomaly over the Pacific and Atlantic sector of the Southern Hemisphere. However, the impacts of tropical heating anomaly in warm winters are weaker than those in La Nina winters because we selected relatively weak ENSO winters. We added the sentence below (page 12, lines 230-231)

“...winters, even without nonlinear process in the LBM.”

Minor points

Given there are nearly twice as many supplementary figures-many of which are important in my opinion-as main figures, perhaps this work would be more suitable in a longer journal that would allow for a more detailed discussion and interpretation of the role of the Tasman Sea compared to the SAM/tropics/zonal wave 3, which are also important.

Thanks for your comment on these other influences. We added the sentence about discussion of other influences. For details, please see reply for your major comments #1.

Your experimental design is not very clear especially with regards to how you are isolating heating in the Tasman Sea.

We added the sentence about details of LBM experiment design to Methods (page 16, lines 316-323).

“To assess the impacts of heating anomaly over the Southern Hemisphere and tropical region, we calculated atmospheric response to heating anomaly over the entire globe [90°S-90°N, 0-360°E], southward of 30°S in the Southern Hemisphere [90°S-30°S, 0-360°E], Pacific sector of Southern Hemisphere [90°S-30°S, 150-330°E] and tropical region [20°S-20°N, 0-360°E]. In addition, guided by the geographical locations of strong atmospheric circulation anomalies, we sub-divided the South Pacific sector into three regions (Drake Passage [90°S-30°S, 150-330°E], Amundsen Sea [90°S-30°S, 150-330°E] and Tasman Sea [90°S-30°S, 150-330°E]).”

As mentioned above, the Tasman Sea warming induced poleward shift of cyclone tracks, resulting in Z300 anomalies between 150 and 210°E (e.g. strong positive Z300 over the east of New Zealand, negative Z300 anomaly east of Antarctica). The northerly warm (southerly cold) advection anomalies associated with positive (negative) Z300 anomalies induced atmospheric heating (cooling) over east of New Zealand (south of Tasman Sea). Therefore, we define the Q1 heating anomaly over these areas (30-90°S, 150-210°E) as effect of Tasman Sea heating.

How did you account for statistical significance in your model results?

Thank you for picking up that we omitted to indicate the important statistical significance of the model results. As mentioned above, each of our experiments was run with 13 different background states, and this sampling allows us to determine whether the model anomalies differ significantly from zero. Figure 5 and supplementary figure 7 now indicate regions over which the differences are significant (95% level). These very strong and organised responses reinforce the arguments made in the paper.

A bit pedantic, but technically Bellingshausen is not on the Antarctic Peninsula, making your title misleading. Do you see similar results for stations located on the Antarctic Peninsula?

Thank you for raising this important point. We have calculated the averaged temperature anomaly at six AP stations (shown in Figure A4a) for June and July, and determined the warm and cold winters of these averages (Figure A4b). The time series is very similar to that in the original submission (just for Bellingshausen) and there are 5 warm events and 6 cold events in common. Analogous to before, we made difference maps of atmospheric and oceanic fields between warm and cold winters at six stations (Figure A5a-c). Not surprisingly the atmospheric and ocean anomalies between warm and cold winters at the six stations are very similar to those between warm and cold winters at the Bellingshausen station. These results mean that the AP warming is influenced by Tasman Sea warming, even temperature anomaly at six stations used as index for warm and cold winters.

Having said this, we prefer to use only the Bellingshausen site for a number of reasons. First, there is a significant amount of missing surface data at several stations as reported by Turner et al. (2016). According to the SCAR READER project site, monthly averaged temperatures are calculated from the 4 daily data at 00, 06, 12 and 18 UTC. The proportion of missing temperature data at the O'Higgins and Rothera stations is more than 40% some Junes and Julys. Some temperature data at the Esperanza station is taken from CLIMAT message, which provide only monthly mean data. Hence, one cannot assess the quality of surface data at the Esperanza station. Second, the Marambio station is coastal station located at a significant elevation (198m), where as the other five stations are coastal low elevation (Esperanza: 13m, O'Higgins: 10m, Faraday/Vernadsky: 11m, Rothera: 32m). In addition, the Faraday and Rothera stations are close to the high mountains in Antarctic Peninsula. The temperatures at these stations are influenced by a warm foehn wind as reported by Kurita et al. (2016). In contrast, the Bellingshausen station is located low level (16m) and is located far from significant topography, meaning that there is little geographical influence on temperature there. In addition, there is almost a complete record of surface temperature data at Bellingshausen (maximum amount of missing data in all the Junes and Julys considered here is only 2%). Therefore, we used temperature at the Bellingshausen station as index for warm and cold winters in this study. It is true that Bellingshausen station is not located in the AP (Fig A4a). However, as we have seen, its temperature is typical of the northern part of the AP. We added the figure and the sentence below (page 11, lines 209-212):

“We have based our analysis on the temperature record at Bellingshausen station, but obtain very similar results using the average temperature anomaly at six Antarctic Peninsula stations (Supplementary Figs. 1a-d, 8c-e).”

And added the sentence below to Methods (pages 13-14, lines 264-272):

“To compare temperature data at the Bellingshausen station with those at five stations (O'Higgins Esperanza, Marambio, Vernadsky and Rothera) in Antarctic Peninsula (Supplementary Fig. 8a), we made the time series of average temperature anomaly at six Antarctic Peninsula stations (Supplementary Fig. 8b). Although the time series of averaged temperature anomaly at six Antarctic Peninsula stations for June and July is very similar to that at Bellingshausen station, however, we used only temperature data at Bellingshausen station, because it has a more complete temperature record. Because the Bellingshausen station has more complete temperature record compared with the other five Antarctic Peninsula stations.”

Figure A4: (a) Elevation map of the Antarctica Peninsula and positions of six station in AP (Bellingshausen, O'Higgins, Esperanza, Marambio, Vernadsky and Rothera). (b) Time series of averaged surface air temperature anomalies ($^{\circ}\text{C}$, deviation from climatology for 1979-2018) during June and July at six stations.

Figure A5: Difference maps for (a) air temperature at 700 hPa ($^{\circ}\text{C}$: shaded) and sea level pressure (hPa: contour), (b) geopotential height at 300 hPa (m: shaded) with the horizontal component of wave-activity flux anomalies (m^2/s^2 : vector) at 300 hPa and (c) sea surface temperature differences between warm and cold winters. Black contours show climate values from 1979 to 2018 for (c) SST. Dotted areas denote significant differences exceeding the 95% confidence level.

What is the primary local mechanism leading to warming at Bellingshausen...is it sea ice anomalies (i.e., local air-sea heat fluxes), thermal advection, something else?

According to results in this study (e.g. Fig.2, S.Fig6), thermal advection associated change in synoptic activity contributes to warming at Bellingshausen station.

What makes June-July so unique that the entire study focuses on only two months? Yes it is true that the mean background circulation is more favorable for Rossby wave propagation out of the tropics during winter, but so too is autumn (Ding and Steig 2013) and spring (Schneider et al. 2012; Clem and Fogt 2015). It isn't clear to me why only two months are presented here.

The greatest warming recorded over the AP occurs in winter. Therefore, we focused on winters in this study.

However, following up your suggestions, we investigated the relationship between change in ocean and AP warming during late winters (July-August) and three winter months (June-August). First, the warm (1985 and 1989) and cold (1980, 1987, 1995, 2007, 2009 and 2011) late winters (July and August) are selected from averaged temperature anomaly at Bellingshausen station (Figure A6a). The tropical SST cooling anomaly causes change in atmospheric circulation over the Southern Hemisphere, contributing to warming over the AP (Figure A6b-d). In contrast, the significant SST warming anomalies are not seen over the Tasman Sea (Figure A6d), meaning that tropical heating anomaly has major role in AP warming.

Second, using the average temperature anomaly at Bellingshausen station during three winter months (Figure A6e), we made the difference maps for atmospheric and oceanic fields between warm and cold three winter months (July and August) (Figure A6f-h). Although tropical SST has significant cooling anomaly, there are no SST anomaly over the Tasman Sea. These results suggested that the Tasman Sea has a minor role in AP warming during late winter (August) when Tasman Sea is colder than it during middle winter (July and June). Therefore, we focused on the relationship between AP warming and Tasman Sea warming during June and July.

Figure A6: (a) Time series of averaged surface air temperature anomalies ($^{\circ}\text{C}$, deviation from climatology for 1979-2018) during July and August at Russian Bellingshausen station. Red and blue dots indicate the top two warm late winters and six cold late winters. Dashed lines show one standard deviation ($\pm 1\sigma$). (b) Air temperature at 700 hPa ($^{\circ}\text{C}$: shaded) and sea level pressure (hPa: contour), (c) geopotential height at 300 hPa (m: shaded) with the horizontal component of wave-activity flux anomalies (m^2/s^2 : vector) at 300 hPa, (d) sea surface temperature differences between warm and cold late winters. Dotted areas denote significant differences exceeding the 95% confidence level. (e-h) Same as (a-d), but for 3-winter months.

I would like to see a time series of Tasman Sea SST alongside Bellingshausen temperature (and other Peninsula station temperature/climate parameter). Has warming of the Tasman Sea overlapped with warming at Bellingshausen? If the relationship is indeed linear, I would expect to see a relationship between warming at Bellingshausen and warming SSTs in the Tasman Sea.

We made time series of observed air temperature at Bellingshausen station and sea surface temperature over the Tasman Sea [30-40°S, 160-175°E] (Figure A7a,b). Figure A7c shows map of correlation coefficient between air temperature at Bellingshausen station and SST. There is significant positive relationship between air temperature at Bellingshausen station and SST over the Tasman Sea.

Figure A7: Time series of (a) air temperature at Bellingshausen station and (b) sea surface temperature (SST) over the Tasman Sea [30-40°S, 160-175°E: green box in (c)] in July and June. (c) Map of correlation coefficient between air temperature at Bellingshausen station and SST. Dotted areas denote significant difference exceeding the 95% confidence level.

References

Clem, K. R., J. A. Renwick, J. McGregor, and R. L. Fogt (2016), *The relative influence of ENSO and SAM on Antarctic Peninsula climate*, *J. Geophys. Res. Atmos.*, 121, 9324–9341, doi:10.1002/2016JD025305.

Clem, K. R., and R. L. Fogt, 2013: *Varying roles of ENSO and SAM on Antarctic Peninsula climate*. *J. Geophys. Res. Atmos.*, doi:10.1002/jgrd.50860

Clem, K. R., and R. L. Fogt, 2015: *South Pacific circulation changes and their connection to the tropics and regional Antarctic warming in austral spring, 1979-2012*. *J. Geophys. Res. Atmos.*, 120, 2773-2792, doi:10.1002/2014JD022940.

Fogt R. L., and D. H. Bromwich, 2006: *Decadal variability of the ENSO teleconnection to the high latitude South Pacific governed by coupling with the Southern Annular Mode*. *J. Climate*, 19, 979-997.

Fogt, R. L., D. H. Bromwich, and K. M. Hines, 2011: *Understanding the SAM influence on the South Pacific ENSO teleconnection*. *Clim. Dyn.*, 36, 1555-1576.

Ding, Q., and E. J. Steig, 2013: *Temperature change on the Antarctic Peninsula linked to the tropical Pacific*. *J. Climate*, 26, 7570-7585, doi:10.1175/JCLI-D-12-00729.1.

Schneider, D. P., C. Deser, and Y. Okumura, 2012a: *An assessment and interpretation of the observed warming of West Antarctica in the austral spring*. *Clim. Dyn.*, 38, 323-347.

L'Heureux, M. L., and D. W. J. Thompson, 2006: Observed relationships between the El-Niño / Southern Oscillation and the extratropical zonal-mean circulation. J. Climate, 19, 276-287.

Turner, J., Lu, H., White, I., King, J. C., Phillips, T., Hosking, J. S., et al. (2016). Absence of 21st century warming on Antarctic Peninsula consistent with natural variability. Nature, 535(7612), 411–415. <https://doi.org/10.1038/nature18645>

Thank you for these very relevant suggestions. We have cited all of these in our revised manuscript.

Reviewer #3 (Remarks to the Author):

This is a generally well written paper and is definitely of publication quality but I don't see anything sufficiently new or substantial here for it to be worthy of a place in a Nature publication. A lot of the analysis of atmospheric changes associated with warm Antarctic Peninsula winters follows a very similar methodology to a paper written more than 20 years ago (Marshall & King 1998 in GRL). While the specific influence of Tasman Sea temperatures in contributing to these changes and the associated model experiments are indeed new, I would have liked to see a greater discussion of how this finding related to other influences on Antarctic climate (other than ENSO and the SAM). For example, the Indian Ocean Dipole is also associated with changing SSTs in the Tasman Sea region and changing SLP in the ABS region of the Southern Ocean (e.g. Nuncio & Yuan 2015 doi: 10.1175/JCLI-D-14-00390.1, Na et al. 2006, doi: 10.1007/BF02842810) but is not mentioned.

Thank you very much for your encouraging comments. Reviewers #1 and #2 raised a very closely related question. Please see the response for comment #1 of reviewers #1 and #2.

We added the sentence about IOD below (page 4, lines 58-63):

“In addition, zonal Indian Ocean SST difference, known as the Indian Ocean Dipole (IOD), has remotely effect on the climate over the Southern Hemisphere^{40,41}. When the eastern Indian Ocean has cold SST anomaly compared with western Indian Ocean (positive phase of IOD) without ENSO, the pressure anomaly is negative (positive) over north (south) of the Ross Sea (Australia), promoting sea ice formation over west of the Ross Sea⁴¹.”

And added the sentence about other influences (ENSO, SAM and IOD) on Antarctic climate to discussion section (pages 11-12, lines 218-242).

Minor points

Line 14. Present tense is incorrect; e.g. Turner et al. (2016) (ref. # 11) show large warming ended at end of 20th Century.

We added “during second half of the 20th century” to abstract.

Line 29. Similar to above, the use of 'since' suggests the dramatic temperature changes are still taking place. Also no explanation of 1957 (IGY?): e.g. there are local temperature records going back further, together with proxy temperatures from ice cores.

We revised the sentence below (page 3, line 29):

“..., from 1958 International Geophysical Year to the late 20th century, ...”

Line 30. I think refs. # 4 and 5 should be switched: e.g. ref. # 4 describes warming from radiosondes, not # 5.

Thank you very much for spotting. This has now been corrected.

Line 33. As currently written, this sentence contradicts the first sentence of this section (e.g. see my previous comments above).

We revised this sentence in first section (page 3, lines 27-29).

Line 38. Marshall & Thompson (2016) doi:10.1002/2015JD024665 is an appropriate reference to add here given their Figure 2 shows heat flux anomalies associated with the SAM and other teleconnection patterns.

Thank you very much. We agree, and have now cited this paper.

Line 38. What do the authors mean by ‘strength’ of the ABSL (now usually called simply ASL in most of the literature)? E.g. the absolute value of the pressure or its value relative to the surrounding region: see discussion in Hosking et al. (ref. # 15).

We replaced “strength” with “sea level pressure (SLP) value”

Line 53. Again see the difference between the PSA1 and PSA2 patterns in Fig. 2 of Marshall & Thompson (2016) as a good example of this.

Thank you very much for the comment. We have cited this paper.

Line 68. Given that the Antarctic Peninsula is one of the few places on the continent when there are actually a reasonable number of stations, it seems strange that the authors only use data from Bellingshausen. For example, they could use the set of six stations that Turner et al. (2016) do, with stations on both the west and east sides of the Peninsula, to provide a proper ‘regional’ set of temperature anomalies. All these stations are coastal low elevation sites. As an example, compare Fig. 1a to Fig. 1 of Marshall & King (1998) in GRL, who used Vernadsky station in an identical way, and the years with SAT more than 1 standard deviation from the mean are not the same (I appreciate that one is JJA and the other is JJ).

Thank you very much for your comment. Reviewer #2 raised a very closely related question. Please see the response for minor comment #4 of reviewer #2.

Line 74. Might be worth saying that CFSR has been validated in the Antarctic (e.g. Bracegirdle & Marshall (2012) doi: 10.1175/JCLI-D-11-00685.1) and found to do a reasonable job.

We cited paper and added the sentence below (page 5, lines 84-85):
“... , which has relatively small biases in atmospheric parameters⁵¹”

Line 134. To me Fig. 4b looks like changes in the split jet (see Bals-Elsholz et al. 2001 in J Clim.) e.g. a strengthening of the polar front jet and a weakening of the subtropical jet in warm winters. A similar result was shown by Marshall & King (1998). Why is the impact on the split jet and its effect on Peninsula SAT not mentioned?

Thank you very much for your comments. We added the sentence below to section (page 8, lines 144-148):
“The zonal wind speed at 300hPa has strongest positive anomaly to the south of New Zealand (Fig. 4b), similar to split jet patterns with strong polar jets^{27,60}. However, in warm winters, the polar jet tends to shift poleward over the region south of Australia and New Zealand (Supplementary Figs. 4, 5b).”

Fig. 5. (b) and (c) should be swapped round to the usual layout of a 2 x 2 figure.

Thank you very much for your suggestion. We replace with remake figure 5.

Antarctic Peninsula warm winters influenced by Tasman Sea temperatures

Kazutoshi Sato^{1*}, Jun Inoue^{2,3}, Ian Simmonds⁴ and Irina Rudeva^{4,5}

¹Kitami Institute of Technology, Kitami, Japan, ²National Institute of Polar Research, Tachikawa, Japan,

³Graduate University for Advanced Studies, SOKENDAI, Hayama Japan, ⁴University of Melbourne,

Melbourne, Australia, ⁵Australian Bureau of Meteorology, Melbourne, Australia

Corresponding author: K. Sato (satokazu@mail.kitami-it.ac.jp)

Abstract

The Antarctic Peninsula of West Antarctica has been one of the most rapidly warming regions on the Earth **during second half of the 20th century**. Changes in the atmospheric circulation associated with the local sea-ice loss and remote tropical climate variabilities have been considered as leading drivers of the change in surface conditions in the region. However, the impacts of climate variabilities over the midlatitudes of the Southern Hemisphere on the Antarctic warming have yet to be quantified. Here, through observation analysis and model experiments, we reveal that increases in winter sea surface temperature (SST) in the Tasman Sea modify the Southern Ocean storm tracks. This, in turn, induces warming over the Antarctic Peninsula and sea ice retreats over the Bellingshausen Sea and Drake Passage via planetary waves triggered in the Tasman Sea. We show that Tasman warm episodes, even in the absence of anomalous tropical SST forcing, deepen the Amundsen-Bellingshausen Seas Low, leading to warm advection over the Antarctic Peninsula.

In recent decades, there is no statistically significant warming trend in the West Antarctica during austral summer due essentially to natural interannual variability¹⁻³. However, from 1958 International Geophysical Year to the late 20th century, the West Antarctic region, and the Antarctic Peninsula in particular, experienced dramatic temperature changes. Significant warming of the troposphere has been revealed in observations (e.g., from surface stations³⁻⁶, radiosondes⁷ and satellites^{8,9}) and reanalysis data^{10,11}. Although positive trends in surface air temperature were found in all seasons, the warming rate of Antarctic Peninsula is particularly marked in winter^{4,9,10}.

The increases in air temperature over Antarctica are related to enhanced warm advection associated with changes in the key atmospheric circulations over the Southern Ocean (e.g., the Amundsen-Bellinghousen Seas Low (ABSL)¹²⁻¹⁶ or the Southern Annular Mode (SAM)¹⁷⁻²¹). The ABSL, the sea level pressure (SLP) value of which is related to the number and intensity of low-pressure systems over the Amundsen-Bellinghousen Seas, plays an important role in the West Antarctic surface climate (e.g., surface air temperature, sea ice extent and wind speed), particularly during winter when cyclones are more numerous and deeper¹⁴⁻¹⁶. The SAM, characterized by westerly circumpolar flow variability associated with the strong meridional pressure gradient between the high and mid-latitudes of the Southern Hemisphere, greatly influences synoptic-scale activity over the Southern Ocean. A positive SAM causes a poleward displacement of the cyclone tracks and reinforces the ABSL, promoting the surface warming over the Antarctic Peninsula^{16,20-22}.

The relationships between atmospheric circulation over the Southern Ocean and tropical oceanic variability, often called “Tropical-polar teleconnections”, have been examined in previous studies^{9,10,16,22-40}. In the austral cold seasons, the heating by anomalous sea surface temperatures (SSTs) in the tropical region generates a Rossby wave train from the tropics to the Antarctic region via the Southern Ocean, influencing its atmospheric circulation variability^{9,10,22-24}. By this mechanism, the El Niño-Southern Oscillation modulates the position and strength of the ABSL^{16,21,22,25,26}. Numerical experiments have shown that the heating anomalies over the tropical region result in the warming (cooling) over the tropical Atlantic and Indian Oceans (eastern Pacific Ocean) and deepened the ABSL and West Antarctic warming²³. In addition, zonal Indian Ocean SST difference, known as the Indian Ocean Dipole (IOD), has remotely effect on the climate over the Southern Hemisphere^{40,41}. When the eastern Indian Ocean has cold SST anomaly compared with western Indian Ocean (positive phase of IOD) without ENSO, the pressure anomaly is negative (positive) over north (south) of the Ross Sea (Australia), promoting sea ice formation over west of the Ross Sea⁴¹.

A recent study has revealed considerable uncertainties in the both pattern and amplitude of the atmospheric response to ENSO due to interannual variability in extratropics⁴². Various studies have reported strong Northern Hemisphere high-latitude atmospheric responses to atmospheric and oceanic forcing over the northern midlatitudes⁴³⁻⁴⁸. However, no previous study has reported the impact of change in SSTs into the Southern Hemisphere midlatitudes on Antarctic climate variability. Here we investigate the linkage

between ocean variabilities in the midlatitudes and the Antarctic warming using a reanalysis dataset and a Linear Baroclinic Model (LBM).

Results

Warm and Cold winters at Bellingshausen station near the Antarctic Peninsula

To understand the connection between interannual variability of SSTs in the midlatitudes and changes in Antarctic Peninsula air temperature, we evaluated a time series of averaged surface air temperature anomalies for June to July over the period 1979-2018 at the Russian Bellingshausen station, which is located just off the northern tip of the Antarctic Peninsula (Fig. 1a, Methods). The data exhibited an interannual standard deviation (σ) of 2.2°C. Six years exceeded $+1\sigma$ ('warm' winters), whereas seven winters recorded mean temperatures less than -1σ ('cold' winters). To investigate the causes of the temperature differences, we constructed difference maps of atmospheric and oceanic fields between composites of warm and the cold winters using the NCEP CFSR reanalysis data^{49,50} (Methods), **which has relatively small biases in atmospheric parameters⁵¹**. A negative sea ice concentration anomaly near the Antarctic Peninsula suggests that this sea ice loss contributed to the warm event (Fig. 1b). However, an east-west seesaw pattern of high and low sea ice concentration anomalies was seen between the Amundsen Sea and near Antarctic Peninsula, resulting from sea ice drifting speed anomalies near the Antarctic Peninsula and over the Amundsen Sea. Therefore, surface conditions (e.g. air temperature, sea ice concentration) in these regions would be induced by atmospheric circulation anomalies over the Southern Ocean.

Atmospheric circulation and Antarctic Peninsula warming

Figure 2a shows the differences in temperature at 700 hPa (T700) and SLP between warm and cold winters. There is a cyclonic anomaly over the Amundsen Sea and an anti-cyclonic anomaly off the east coast of Argentina. This pattern resulted in a strong SLP gradient over the Drake Passage, leading to northerly warm advection over the Weddell Sea and Antarctic Peninsula. The poleward drifting of sea ice associated with the northerly wind anomaly decreased sea ice extents over the Bellingshausen Sea and Drake Passage (Fig. 1b). The strong Amundsen Sea cyclonic anomaly, which enhanced the warm advection over the Antarctic Peninsula, has also been referred to as the ABSL in some studies^{18,19}. In contrast, the southerly cold advection from Antarctica induced cold temperature anomalies and equatorward shift of sea ice over the Ross and Amundsen Seas.

The 300 hPa geopotential height (Z300) anomalies show a strong wave-like structure excited over the Southern Ocean (Fig. 2b), **resembling a zonal wave 3 pattern⁵²**. This pattern is similar to atmospheric responses to tropical SST anomalies reported by previous studies (e.g., **Tropical-polar teleconnections^{10,11,22-24}**) (Supplementary Fig. 1), although the strongest wave anomaly pattern appears over the mid and high latitudes. The propagation of wave activity flux as indicated by arrows⁵³ shows a wave train originating from south of the New Zealand region. This suggests that these wave-like anomaly patterns are associated

with the midlatitudes variability, particularly changes in SST in the Tasman Sea. In addition, there was a positive relationship between SST in the Tasman Sea and T700 over the Antarctic Peninsula (Supplementary Fig. 2).

To address the causes of SLP anomalies over the Southern Ocean and Antarctica, we show the composite maps of the densities of cyclones in warm and cold winters (Fig. 3a, Methods). An area of strong cyclone activity was found around the Antarctic coastline in both composites, particularly in the Indian sector (Supplementary Fig. 3). These patterns are consistent with the results of previous studies¹⁶⁻²². However, there were positive differences in the density of cyclones over the Antarctic coastal regions (Fig. 3a). In contrast, negative differences were seen over the Drake Passage and in the central Pacific to the east of New Zealand. This pattern suggests that the cyclones tend to shift poleward in warm winters. In addition, positive precipitation anomalies over the south-eastern Pacific and negative anomalies to the east of Argentina and New Zealand are consistent with the geographical distribution of changes in cyclone frequency (Fig. 3b).

Impact of Tasman Sea warming on the atmospheric circulation over the Southern Ocean

In recent years, considerable warming has occurred in the Tasman Sea⁵⁴, and the frequency and duration of marine heatwaves have increased^{55,56}. Satellite data show a change in the South Pacific subtropical gyre associated with enhanced wind stress curl leading to recent SST warming in the Tasman Sea⁵⁷. In addition, the warm East Australian Current in the west Tasman Sea has exhibited a poleward shift and intensification in recent times⁵⁸. The midlatitude and polar teleconnection patterns are seen between the Tasman Sea and Antarctica⁵⁹. **Therefore, a warming Tasman Sea has strengthened the SST gradient between high and mid latitudes (increased baroclinicity), and hence has influenced on cyclone tracks over the Southern Ocean.**

To investigate SST anomalies in the Tasman Sea, we focus on the difference in SST between Bellingshausen warm and cold winters on the Antarctic Peninsula (Fig. 4a, Supplementary Fig. 5a). A remarkable feature of Fig. 4a is that the SST differences in the Southern Ocean are quite small, with the notable exception of the significant SST differences in the Tasman Sea and to the east of New Zealand. **The zonal wind speed at 300hPa has strongest positive anomaly to the south of New Zealand (Fig. 4b), similar to split jet patterns with strong polar jets^{27,60}. However, in warm winters, the polar jet tends to shift poleward over the region south of Australia and New Zealand (Supplementary Figs. 4, 5b).** In warm winters, higher SST in the Tasman Sea strengthens the meridional SST gradient between the Tasman Sea and Southern Ocean (Supplementary Fig. 5c). Therefore, the upper level jet shifts poleward over the south of the Tasman Sea (Fig. 4b, Supplementary Fig. 5b). The upper level jet anomaly shifted cyclone tracks to the south over the Tasman Sea, **causing change in the atmospheric heating in the downstream regions (i.e. Pacific sector of the Southern Ocean).**

Atmospheric response to the forcing over the midlatitudes in the western Pacific

To address the atmospheric response to the positive SST anomaly over the Tasman Sea during warm and cold winters from a theoretical point of view, we calculated a steady atmospheric response using the LBM (Methods). Figure 5a shows the vertical-averaged (from sigma levels 1.0-0.3) difference in the diabatic heating rate (Q1), which is made up of three terms (local tendency, horizontal advection, and vertical advection of potential temperature) reflecting radiative effects, and latent and sensible heating⁶¹, between warm and cold winters. The total Q1 anomaly pattern over the Southern Ocean was similar to its horizontal advection component (Fig. 5a and Supplementary Fig. 6b), indicating that the major source for Q1 was the difference in horizontal advection heating associated with change in cyclone tracks. A feature of the Z300 response when driving the LBM with the Q1 anomaly over the entire globe was a positive anomaly of Z300 east of Argentina and a negative anomaly over Amundsen Sea, resulting in anomalous northerly warm advection in the lower troposphere over West Antarctica (Fig. 5b). This anomalous pattern is similar to that observed, although the amplitudes of Z300 are smaller than those exhibited by the CFSR reanalysis (Fig. 2b).

To assess the impacts of the removal tropical (and Northern Hemisphere) atmospheric anomalies on atmospheric circulations in the Southern Hemisphere, we conducted a further experiment using a heating anomaly poleward of 30°S only (area enclosed by purple dashed lines in Fig. 5c, Methods). A remarkable aspect of the response is the strong similarity of Figs. 5b and 5c. They reveal that, when comparing the anomalous circulations associated with warm and cool Bellingshausen winters, are dominated by changes and forcings in the southern extratropics, and tropical influences are minimal. In fact, when weak Q1 anomaly forced over the tropical regions (20°S-20°N, 150°-330°E), there is no strong Z300 response over the Southern Ocean (Supplementary Fig. 7a).

This surprising result led us to conduct an experiment in which the Q1 anomalies were restricted to the ‘half hemisphere’ of the Pacific and western Atlantic sectors (90°S-30°S, 150°-330°E: enclosed by purple dashed lines in Fig. 5d). This resulted in a very similar anomalous Z300 pattern, albeit a little muted, over the forcing sector, including a strong negative anomaly over the Amundsen Sea. To shed additional light on these responses we ran four further experiments by restricting the Q1 forcing areas to the Amundsen Sea (90°S-30°S, 210°-270°E), Drake Passage (90°S-30°S, 270°-330°E) and Tasman Sea (90°S-30°S, 150°-210°E) (Supplementary Fig. 7b-d, Methods). **The positive Z300 anomaly over the east of New Zealand and negative Z300 anomaly over the south of New Zealand and Australia are induced by poleward shift of cyclone tracks related to Tasman Sea warming, leading to atmospheric heating anomalies over the Tasman Sea sector (Fig. 2b, Supplementary Fig. 7b). The atmospheric response to the heating anomaly over the Tasman Sea sector includes a strong ridge across the Pacific at 45-55°S and trough over the Amundsen Sea (Supplementary Fig. 7b). This ridge is reminiscent of that in the global-forcing experiment (Fig 5b), but is not seen in any of the other sector experiments shown in Supplementary Fig. 7. In contrast, the amplitude of trough is smaller than that in the Amundsen Sea-forcing experiment (Supplementary Fig. 7b,c). The strongest negative Z300 responses, which induced Antarctic Peninsula warming, came from Q1 cooling over the Amundsen sector (Supplementary Fig. 7c). However, relatively weak trough associated with Tasman Sea**

forcing causes southerly cold advection from Antarctica, resulting in cooling anomaly over the Amundsen Sea (Supplementary Fig. 7b). These experiments indicate that the Antarctic Peninsula warming is triggered by the Tasman Sea warming.

Discussion

In summary, we have shown that warm winter episodes in the Tasman Sea influence warm temperature anomalies over key regions of West Antarctica including Antarctic Peninsula, through a poleward shift of the South Pacific cyclone track. We have based our analysis on the temperature record at Bellingshausen station, but obtain very similar results using the average temperature anomaly at six Antarctic Peninsula stations (Supplementary Figs. 1a-d, 8c-e). The LBM results present new insights into remote atmospheric responses to Southern Hemisphere midlatitude perturbations and the role that these may play in influencing this sensitive region of West Antarctica. This broad concept is consistent, though in another context, with the analysis that showed that Southern Hemisphere synoptic perturbations in the midlatitudes tend to lead variations in the Hadley Cell width⁶².

Previous investigations have shown that strengthening of the ABSL can be induced by the La Niña, positive SAM and negative IOD²²⁻⁴⁰. In warm AP winters, the wave activity flux anomalies appear to originate from subtropical Pacific to Antarctica, suggesting that the central Pacific cooling influences atmospheric circulation over the Southern Hemisphere as reported by previous studies. In the La Niña winters, the amplitudes and areas of tropical Pacific SST cooling are larger than those in warm winters (Methods, Supplementary Fig. 1h). These strong cooling anomalies contribute to the clear wave-like anomalies from the tropical Pacific region to West Antarctica (Supplementary Fig. 1e-h). In addition, in warm AP winters without strong El Niño, there are no negative tropical SST anomaly and wave activity flux from Pacific subtropical to West Antarctic (Methods, Supplementary Fig. 1q-t). Therefore, the amplitude of atmospheric response to tropical Pacific cooling during warm AP winters would be weaker than that in strong La Niña winters, even without nonlinear process in the LBM. From these results, the tropical heating has an important role in Antarctic Peninsula warming. However, when tropical heating is weak, the Antarctic Peninsula warming is influenced by the Tasman Sea warming. In positive (stronger) SAM winters, the atmospheric circulation anomalies resemble those in warm winters over the Antarctic Peninsula (Supplementary Fig. 1i-l). The northeast Antarctic Peninsula warming, which has significant positive correlation with SAM³³, is seen clearly in strong positive SAM winters. However, in warm AP winters without strong SAM, the wave activity flux remains strong from south of New Zealand to west Antarctic (Supplementary Fig. 1u-x). In contrast, in negative IOD winters, there are no clear wave-like anomalies over the Southern Hemisphere (Supplementary Fig. 1m-p), because atmospheric anomalies over the Southern Hemisphere linked to the IOD are found during late winter and early spring^{40,41}.

In La Nina and positive SAM winters, the atmospheric circulation anomalies have relatively small amplitudes (Supplementary Fig. 1e,f,i,j). The SST anomalies in the Tasman Sea are absent in La Niña and

positive SAM winters (Supplementary Fig. 1h,l). These results suggest that the Tasman Sea warming induced a positive SAM-like pattern in the troposphere with a Z300 dipole over the Southern Hemisphere. Our study has particular relevance as the Tasman Sea has been identified as a global warming hotspot and one that will be subject to dramatic increases in marine heat waves^{53,54,63} with potentially significant impact on Antarctic Peninsula climate.

Methods

Observation data. The Antarctic Climate Data, which includes temperature, surface pressure, and winds observed at manned and automatic weather stations, are available from the SCAR READER project (<https://legacy.bas.ac.uk/met/READER>)⁶⁴. In this study, the Russian Bellingshausen station was chosen to avoid the effects on temperature of elevations in Antarctica. In addition, the station, which located in a marine environment, can reflect internal ocean variability. Monthly averaged surface temperature data from Bellingshausen station were used in Fig. 1a. From this time series, we selected typical warm and cold winters (June to July) for which the magnitude of the temperature anomaly values exceeded one standard deviation (warm winters: 1989, 1998, 2000, 2004, 2010, and 2016; cold winters: 1980, 1987, 1991, 1992, 1995, 2007, and 2011).

To compare temperature data at the Bellingshausen station with those at five stations (O'Higgins Esperanza, Marambio, Vernadsky and Rothera) in Antarctic Peninsula (Supplementary Fig. 8a), we made the time series of average temperature anomaly at six Antarctic Peninsula stations (Supplementary Fig. 8b). Although the time series of averaged temperature anomaly at six Antarctic Peninsula stations for June and July is very similar to that at Bellingshausen station, however, we used only temperature data at Bellingshausen station, because it has a more complete temperature record. Because the Bellingshausen station has more complete temperature record compared with the other five Antarctic Peninsula stations.

Reanalysis data. We used 6-hourly data of the Climate Forecast System Reanalysis⁴⁹ (CFSR) from January 1979 to March 2011 and CFS Version2⁵⁰ from April 2011. The CFSR and CFS data on a 0.5° horizontal grid are provided by the National Centers for Environmental Prediction (NCEP). We used meteorological (SLP, air temperature, wind speed, geopotential height and precipitation) and oceanographic (sea ice concentration, sea surface temperature, and ice drifting speed) parameters.

It is important to establish that our results do not depend on this specific choice of reanalysis set. Accordingly, we also make use of the European Centre for Medium-Range Weather Forecasts (ECMWF) ERA5 compilation⁶⁵ with horizontal resolution of 0.5°×0.5°. We constructed difference maps of T700, SLP, Z300 and wave activity fluxes at 300hPa (WAF300) with the ERA5 data in the same form as Fig. 2 (Supplementary Fig. 9). To reduce uncertainty in reanalysis data, the mean of the 10 ensemble members of ERA5 were used. The plots differed little from those constructed with the NCEP product.

Cyclone tracking data. We used the cyclone tracking data derived from an algorithm developed at the University of Melbourne⁶⁶. This algorithm identifies the cyclone positions at 6-hourly intervals based on the Laplacian of pressure at each grid. Cyclones lasting less than 24 hourly were removed from the analysis. To investigate cyclone density, we calculated the densities of cyclones a circular grid⁶⁷ with circular cells of a 5° latitude radius (equivalent to approximately 308,025 km²) for each year.

Linear Baroclinic Model. A linear baroclinic model (LBM)⁶⁸ was used to investigate the steady atmospheric response to a heating anomaly over a target area. The LBM was based on the primitive equations linearised about a basic state on a sphere, at resolution of T42L11. Because of the simplified framework of LBMs, where nonlinear processes in the atmosphere are removed, the results are much more easily interpreted. In the present study, the model is linearised about 13 different background states for various June and July mean conditions, derived from the CFSR reanalysis dataset (Supplementary Table 1). We calculated the atmospheric responses for each of these states and show the ensemble mean of these simulations.

The diabatic heating rate data, Q1 estimated through a heat budget analysis based on the thermodynamic equations⁶¹, was calculated by the CFSR. The Q1 reflects the sensible and latent heating and radiation, and was used as a steady forcing for the LBM. Q1 is defined by:

$$Q1 = \left(\frac{p}{p_0}\right)^\kappa \left(\frac{\partial \bar{\theta}}{\partial t} + \bar{V} \cdot \nabla \bar{\theta} + \frac{\partial \bar{\omega} \bar{\theta}}{\partial p} \right)$$

where p is the pressure, $p_0=1000$ hPa, V is the horizontal velocity, θ is the potential temperature, and ω is the vertical p -velocity, and the overbar refers to the winter time averages. In the equation $\kappa = R/c_p$, R is the gas constant and c_p is the specific heat of air at constant pressure. The three terms on the right hand side show the changes associated with local tendency, horizontal advection, and vertical advection, respectively. Supplementary Fig. 6 shows the vertical averaged difference in each term of the Q1 between warm and cold winters.

To assess the impacts of heating anomaly over the Southern Hemisphere and tropical region, we calculated atmospheric response to heating anomaly over the entire globe [90°S-90°N, 0-360°E], southward of 30°S in the Southern Hemisphere [90°S-30°S, 0-360°E], Pacific sector of Southern Hemisphere [90°S-30°S, 150-330°E] and tropical region [20°S-20°N, 0-360°E]. In addition, guided by the geographical locations of strong atmospheric circulation anomalies, we sub-divided the South Pacific sector into three regions (Drake Passage [90°S-30°S, 150-330°E], Amundsen Sea [90°S-30°S, 150-330°E] and Tasman Sea [90°S-30°S, 150-330°E]).

NINO3.4 index. Monthly averaged SST anomalies were calculated for the NINO3.4 region (5°S-5°N, 190°-240°E). For our June to July period (Supplementary Fig. 10a), analogous to above, we determine La Niña and El Niño winters for which the NINO3.4 index values exceeded one standard deviation (La Niña winters:

1984, 1985, 1988, 1998, 1999, and 2010; El Niño winters: 1987, 1997, 2002, 2009, and 2015). Using the CFSR reanalysis data, we made difference maps of T700 with SLP, Z300 with WAF300, and SST with climatology SST values between La Niña and El Niño winters, such as Fig. 2 (Supplementary Fig. 1e-h).

Southern Annular Mode (SAM) index. Observation-based Southern Hemisphere Annular Mode Index data⁶⁹ were used as the SAM index (available from <https://legacy.bas.ac.uk/met/gjma/sam.html>). From this anomaly time series for June to July (Supplementary Fig. 10b), we extract strong positive and negative SAM winters, defined as those for which the magnitude of the SAM index values exceeded one standard deviation (SAM+ winters: 1979, 1989, 2004, 2006, 2010, 2012, 2015, and 2016; SAM- winters: 1988, 1990, 1992, 1994-95, and 2007). Using the CFSR reanalysis data, we made difference maps of T700 with SLP, Z300 with WAF300 and SST with climatology SST values between SAM+ and SAM- winters such as Fig. 2 (Supplementary Fig. 1i-l).

Indian Ocean Dipole index. The Indian Ocean Dipole (IOD) index⁷⁰ is defined as the SST anomaly difference between tropical western Indian Ocean (10°S-10°N, 50°-70°E) and tropical southeast Indian Ocean (10°S-eq, 90°-110°E) (Supplementary Fig. 10c). From this anomaly time series for June to July, we selected strong positive and negative IOD winters for which the IOD index values exceeded one standard deviation (IOD+ winters: 1982-83, 1994, 1997, 2003, 2008, 2015, 2017, and 2018; IOD- winters: 1992, 1996, 2002, 2004, and 2016). Using the CFSR reanalysis data, we made difference maps of T700 with SLP, Z300 with WAF300, and SST with climatology SST values between IOD+ and IOD- winters such as Fig. 2 (Supplementary Fig. 1m-p).

Removal of the impacts of ENSO and SAM. To remove the impact of ENSO in warm and cold winters, we composited the atmospheric and ocean fields for warm and for cold AP winters without strong ENSO winters (e.g. strong La Niña [1998, 2010] and El Niño [1987] winters) (Supplementary Fig. 10d). We then formed the difference maps of these (Supplementary Fig. 1q-t). In a similar fashion we removed the impact of SAM on warm and cold AP winters by assembling the difference maps of atmosphere and ocean between warm and cold AP winters without strong SAM (e.g., SAM+ [1989, 2004, 2010, 2016] and SAM- (1992, 1995, 2007) winters) (Supplementary Figs. 1u-x, 10e).

Data availability

The surface temperature datasets were obtained from the SCAR READER project (<https://legacy.bas.ac.uk/met/READER>). We used the observation-based Southern Hemisphere Annular Mode Index (available from <https://legacy.bas.ac.uk/met/gjma/sam.html>). To composite atmospheric and oceanic maps, we used the CFSR and ERA5 reanalyses provided by the ECMWF and NCEP.

Code availability

All codes used to analyze and plot the data are available from the corresponding author on request.

References

1. Turner, J. et al. Absence of 21st century warming on Antarctic Peninsula consistent with natural variability. *Nature*, 535, 411–415, doi:10.1038/nature1864 (2016).
2. Nicolas, J. P. & Bromwich, D. H. New reconstruction of Antarctic near-surface temperatures: Multidecadal trends and reliability of global reanalyses. *J. Clim.*, 27, 8070-8093, doi: 10.1175/JCLI-D-13-00733.1 (2014).
3. Gonzalez, S. & Fortuny, D. How robust are the temperature trends on the Antarctic Peninsula? *Antarctic Science*, 30, 322-328, doi: 10.1017/s0954102018000251 (2018).
4. Turner, J., Lachlan-Cope, T. A., Colwell, S., Marshall, G. J. & Connolley, W. M. Significant warming of the Antarctic winter troposphere. *Science*, 311, 1914-1917 (2006).
5. Bromwich, D. H., Nicolas, J. P., Monaghan, A. J., Lazzara, M. A., Keller, L. M., Weidner, G. A. & Wilson, A. B. Central West Antarctica among the most rapidly warming regions on Earth. *Nat. Geosci.*, 6(2), 139– 145 (2013).
6. Steig, E. J., Schneider, D. P., Rutherford, S. D., Mann, M. E., Comiso, J. C. & Shindell, D. T. Warming of the Antarctic ice-sheet surface since the 1957 International Geophysical Year. *Nature*, 457, 459–462, doi:10.1038/nature07669 (2009).
7. Screen, J. A. & Simmonds, I. Half-century air temperature change above Antarctica: Observed trends and spatial reconstructions. *J. Geophys. Res.*, 117, D16108, doi:10.1029/2012JD017885 (2012).
8. O'Donnell, R., Lewis, N., McIntyre, S. & Condon, J. Improved methods for PCA-based reconstructions: Case study using the Steig et al. (2009) Antarctic temperature reconstruction. *J. Clim.*, 24, 2099-2115 (2010).
9. Ding, Q., Steig, E. J., Battisti, D. S. & Kuttel, M. Winter warming in West Antarctica caused by central tropical Pacific warming. *Nat. Geosci.*, 4, 398-403 (2011).
10. Ding, Q. & Steig, E. J. Temperature change on the Antarctic Peninsula linked to the tropical Pacific. *J. Clim.*, 26, 7570–7585. doi: 10.1175/JCLI-D-12-00729.1 (2013).
11. Turner, J., Maksym, T., Phillips, T., Marshall, G. J. & Meredith, M. P. The impacts of changes in sea ice advance on the large winter warming on the western Antarctic Peninsula. *Int. J. Climatol.*, 33, 852–861, doi:10.1002/joc.3474 (2013).
12. Simmonds, I., Keay, K. and Lim, E. P. Synoptic activity in the seas around Antarctica. *Mon. Weather Rev.*, 131(2), 272–288, doi:10.1175/1520-0493(2003)131 (2002).
13. Simmonds, I. & Keay, K. Mean Southern Hemisphere extratropical cyclone behavior in the 40-year NCEP-NCAR reanalysis. *J. Clim.*, 13, 873–885 (2000).
14. Fogt, R. L., Wovrosh, A. J., Langen, R. A., & Simmonds, I. The characteristic variability and connection to the underlying synoptic activity of the Amundsen-Bellinghousen Seas Low. *J. Geophys. Res.*, 117, D07111, doi:10.1029/2011JD017337 (2012).
15. Hosking, J. S., Orr, A., Marshall, G. J., Turner, J. & Phillips, T. The influence of the Amundsen–Bellingshausen Seas low on the climate of West Antarctica and its representation in coupled climate model simulations. *J. Clim.*, 26, 6633– 6648, doi:10.1175/JCLI-D-12-00813.1 (2013).
16. Clem, K.R., Renwick, J.A. & McGregor, J. Large-scale forcing of the Amundsen Sea low and its influence on sea ice and West Antarctic temperature. *J. Clim.*, 30(20), 8405– 8424 (2017).
17. Thompson, D. W. J. & Solomon, S. Interpretation of recent Southern Hemisphere climate change. *Science*, 296, 895-899 (2002).
18. Simmonds, I. Modes of atmospheric variability over the Southern Ocean. *J. Geophys. Res.*, 108(C4), 8078, doi:10.1029/2000JC000542 (2003).
19. Thompson, D. W. J., Solomon, S., Kushner P. J., England M. H., Grise, K. M. & Karoly D. J. Signatures of the Antarctic ozone hole in Southern Hemisphere surface climate change, *Nat. Geo.*, 4, 741-749 (2011).
20. Bader, J., Flügge, M., Kvamstø, N. G., Mesquita, M. D. S. & Voigt A. Atmospheric winter response to a projected future Antarctic sea-ice reduction: a dynamical analysis. *Clim Dyn*, 40, 2707-2718. <https://doi.org/10.1007/s00382-012-1507-9> (2013).
21. Simmonds, I. & King, J. C. Global and hemispheric climate variations affecting the Southern Ocean. *Antarc. Sci.*, 16, 401-413, doi: 10.1017/S0954102004002226 (2004)
22. Ciasto, L. M., Simpkins, G. R., and England, M. H. Teleconnections between tropical Pacific SST anomalies and extratropical Southern Hemisphere climate. *J. Clim.*, 28(1), 56– 65, doi: 10.1175/JCLI-D-14-00438.1 (2015).
23. Li, X., Holland, D., Gerber, P., and Yoo, C. Rossby waves mediate impacts of tropical oceans on West Antarctic atmospheric circulation in austral winter. *J. Clim.*, 28, 8151– 8164, doi: 10.1175/JCLI-D-15-0113.1 (2015).
24. Irving, D. & Simmonds, I. A new method for identifying the Pacific–South American pattern and its influence on regional climate variability. *J. Clim.*, 29, 6109–6125. Doi:10.1175/JCLI-D-15-0843.1 (2016).
25. Pezza, A. B., Rashid, H. A. & Simmonds, I. Climate links and recent extremes in Antarctic sea ice, high-latitude cyclones, Southern Annular Mode and ENSO. *Climate Dyn.*, 38, 57-73, doi: 10.1007/s00382-011-1044-y (2012).

26. Stammerjohn, S. E., Martinson, D. G., Smith, R. C., Yuan, X. & Rind, D. Trends in Antarctic annual sea ice retreat and advance and their relation to ENSO and southern annular mode variability. *J. Geophys. Res.*, 113, C03S90, doi:10.1029/2007JC004269 (2008).
27. Marshall J. G. & King, C. J. Southern Hemisphere circulation anomalies associated with extreme Antarctic Peninsula winter temperature. *Geophys. Res. Lett.* 25, 2437–2440 (1998).
28. Ding, Q., Steig, J. E., Battisti, S. D. & Wallace, M. J. Influence of the Tropics on the Southern Annular Mode. *J. Clim.* 25, 6330–6348 (2012).
29. Fogt, L. R. & Bromwich, H. D. Decadal variability of the ENSO teleconnection to the high-latitude south Pacific governed by coupling with the Southern Annular Mode. *J. Clim.* 19, 979–997 (2005).
30. Clem, R. K. & Fogt, L. R. South Pacific circulation changes and their connection to the tropics and regional Antarctic warming in austral spring, 1979–2012 *J. Geophys. Res. Atmos.*, 120, 2773–2792 (2015).
31. Marshall, J. G. & Thompson J. W. The signatures of large-scale patterns of atmospheric variability in Antarctic surface temperature. *J. Geophys. Res. Atmos.*, 121, 3276–3289, doi:10.1002/2015JD024665 (2016).
32. Clem, R. K. & Fogt, L. R. Varying roles of ENSO and SAM on the Antarctic Peninsula climate in austral spring. *J. Geophys. Res. Atmos.*, 118, 11481–11492, doi:10.1002/jgrd.50860 (2013).
33. Clem, R. K., Renwick, A. J., McGregor, J. & Fogt, L. R. The relative influence of ENSO and SAM on Antarctica Peninsula climate. *J. Geophys. Res. Atmos.*, 121, 9324–9341, doi:10.1175/2016JD025305 (2016).
34. L’Heureux, L. M. & Thompson, W. J. D. Observed relationships between the El Niño-Southern Oscillation and the extratropical zonal-mean circulation. *J. Clim.*, 19, 276–287 (2005).
35. Li, X., Holland, D. M., Gerber, E. P. & Yoo, C. Impacts of the north and tropical Atlantic Ocean on the Antarctic Peninsula and sea ice. *Nature*, 505, 538–542, doi:10.1038/nature12945 (2014).
36. Irving, D. & Simmonds, I. A novel approach to diagnosing Southern Hemisphere planetary wave activity and its influence on regional climate variability. *J. Clim.*, 28, 9041–9057, doi: 10.1175/JCLI-D-15-0287.1 (2015).
37. Yiu, Y. Y. S. & Maycock, A. C. On the seasonality of the El Niño teleconnection to the Amundsen Sea region. *J. Climate*, 32, 4829–4845, doi: 10.1175/JCLI-D-18-0813.1 (2019).
38. Schneider, P. D., Deser, C. & Okumura, Y. An assessment and interpretation of the observed warming of West Antarctica in the austral spring. *Clim Dyn*, 38, 323–347. Doi: 10.1007/s00382-010-0985-x (2011).
39. Fogt, L. R., Bromwich, H. D. and Hines, M. K. Understanding the SAM influence on the South Pacific ENSO teleconnection, *Clim Dyn*, 36, 1555–1576 (2010).
40. Liu, N., Jia, Z., Chen, H. Hua, F. & Li, Y. Southern high latitude climate anomalies associated with the Indian Ocean dipole mode. *Chin. J. Ocean. Limnol.*, 24:2, 125–128, DOI: 10.1007/BF02842810 (2005).
41. Nuncio, M. & Yuan, X. The influence of the Indian Ocean dipole on Antarctic sea ice. *J. Clim.* 28, 2682–2690 (2015).
42. Deser, C., Simpson, R. I., Mckinnon, A. K. & Phillips, S. A. The Northern Hemisphere extratropical atmospheric circulation response to ENSO: How well do we know it and how do we evaluate models accordingly. *J. Clim.*, 30, 5059–5082. doi: 10.1175/JCLI-D-18-0782.1 (2017).
43. Screen, J. A. & Simmonds, I. Exploring links between Arctic amplification and mid-latitude weather. *Geophys. Res. Lett.*, 40, 959–964 (2013).
44. Kosaka, Y. & Xie, S.-P. Recent global-warming hiatus tied to equatorial Pacific surface cooling. *Nature*, 501, 403–407. Doi:10.1038/nature12534 (2013).
45. Sato, K, Inoue, J. & Watanabe, M. Influence of the Gulf Stream on the Barents Sea ice retreat and Eurasian coldness during early winter, *Environ. Res. Lett.*, 9, 084009 (2014).
46. Simmonds, I. and Govekar, P. D. What are the physical links between Arctic sea ice loss and Eurasian winter climate?, *Environ. Res. Lett.*, 9, 101003, doi:10.1088/1748-9326/9/10/101003 (2014).
47. Nakanowatari, T., Inoue, J., Sato, K. & Kikuchi, T. Summertime atmosphere-ocean preconditionings for the Bering Sea ice retreat and the following severe winters in North America, *Environ. Res. Lett.*, 10, 094023 (2015).
48. Tokinaga, H., Xie, S.-P. & Mukougawa, H. Early 20th-century Arctic warming intensified by Pacific and Atlantic multidecadal variability. *PNAS*, 114(24), 6227–6232. Doi:10.1073/pnas.1615880114 (2017).
49. Saha, S. et al. The NCEP climate forecast system reanalysis. *Bull. Am. Meteorol. Soc.*, 91, 2185–2208. (2010).
50. Saha, S. et al. The NCEP climate forecast system version 2. *J. Clim.*, 27, 2185–2208 (2014).
51. Bracegirdle, J. T. & Marshall, G. J. The reliability of Antarctic tropospheric pressure and temperature in the Latest Global Reanalysis. *J. Clim.*, 25, 7138–7146, doi:10.1175/JCLI-D-11-00685.1 (2012).
52. Raphael, M. N. A zonal wave 3 index for the Southern Hemisphere. *Geophys. Res. Lett.* 31, L23212 (2004).
53. Takaya, K. & Nakamura, H. A formulation of a phase-independent wave-activity flux for stationary and migratory quasigeostrophic eddies on a zonally varying basic flow. *J. Atmos. Sci.*, 58, 608–627 (2001).
54. Oliver, E. C. J., Benthuisen J. A., Bindoff, N. L., Hobday, A. J., Holbrook, N. J., Mundy, C. N. and Perkins-Kirkpatrick, S. E. The unprecedented 2015/16 Tasman Sea marine heatwave. *Nat. Commun.*, 8, 16101, doi: 10.1038/ncomms16101 (2017).

55. Oliver, E. C. J., Wotherspoon, S. J., Chamberlain, M. A. & Holbrook, N. J. Projected Tasman Sea extremes in sea surface temperature through the twenty-first century. *J. Clim.*, 27, 1980–1998, doi:10.1175/JCLI-D-13-00259.1 (2014).
56. Oliver, E. C. J. et al. Longer and more frequent marine heatwaves over the past century. *Nat. Commun.*, 9, 03732, doi: 10.1038/s41467-018-03732-9 (2018).
57. Shears, N. T. & Bowen, M. M. Half a century of coastal temperature records reveal complex warming trends in western boundary currents. *Scientific reports*, 7, 14527. Doi:10.1038/s41598-017-14944-2 (2017).
58. Wu, L. et al. Enhanced warming over the global subtropical western boundary currents. *Nat. Clim. Change*, 2, 161-166 (2012).
59. Liess, S., Kumar, A., Snyder, K. P., Kawale, J., Steinhäuser, K., Semazzi, H. M. F., Ganguly, R. A., Samatova, F. N. & Kumar, V. Different modes of variability over the Tasman Sea: Implications for regional climate. *J. Clim.*, 27, 8466–8486 (2014).
60. Bals-Elsholz, T. M., Atallah E. H., Bosart, L. F., Wasula, A. T., Cempa J. M. & Lupo, R. A. The wintertime Southern Hemisphere split jet: Structure, variability, and evolution. *J. Clim.* 14, 4191–4215 (2001).
61. Yanai, M., Esbensen, S. & Chu, J. H. Determination of bulk properties of tropical cloud clusters from large-scale heat and moisture budgets. *J. Atmos. Sci.*, 30, 611-627 (1973).
62. Rudeva, I., Simmonds, I., Crock, D. & Bosch, G. Midlatitude fronts and variability in the Southern Hemisphere tropical width. *J. Clim.*, 32, 8243-8260. doi: 10.1175/JCLI-D-18-0782.1 (2019).
63. Kirtman, B. et al. Near-term climate change: Projections and predictability. *Climate Change 2013: The Physical Science Basis. Contribution of Working Group I to the Fifth Assessment Report of the Intergovernmental Panel on Climate Change*. T. F. Stocker et al., Eds., Cambridge University Press, 953-1028, (2013).
64. Turner, J. et al. The SCAR READER project: toward a high-quality database of mean Antarctic meteorological observations, *J. Clim.*, 17, 2890-2898 (2004).
65. Hersbach, H. et al. Operational global reanalysis: progress, future directions and synergies with NWP. *ERA Report Series*, 27, 1-63. (2018) <https://www.ecmwf.int/en/elibrary/18765-operational-global-reanalysis-progress-future-directions-and-synergies-nwp>
66. Simmonds, I. & Murray, R. J. Southern extratropical cyclone behaviour in ECMWF analyses during the FROST special observing periods. *Wea. Forecasting*, 14, 878–891 (1999).
67. Rudeva, I., Gulev, S. K., Simmonds, I. & Tilinina, N. The sensitivity of characteristics of cyclone activity to identification procedures in tracking algorithms. *Tellus A: Dynamic Meteorology and Oceanography*, 66:1, 24961, DOI: 10.3402/tellusa.v66.24961 (2014).
68. Watanabe, M. & Kimoto, M. Atmosphere-ocean thermal coupling in the Northern Atlantic: a positive feedback. *Q. J. R. Meteorol. Soc.*, 126, 3343-3369 (2000).
69. Marshall, G. J. Trends in the Southern Annular Mode from observations and reanalyses. *J. Clim.* 16, 4134–4143 (2003).
70. Saja, N. H., Goswami, B. N., Vinayachandran, P. N. & Yamagata, T. A. A dipole mode in the tropical Indian Ocean. *Nature*, 401, 360-363 (1999).

Acknowledgments

We would like to thank Prof. M. Watanabe, who provided the LBM for this study. This work was supported by a JSPS Overseas Research Fellowship, JSPS KAKENHI (19K14802, 18H05053). Parts of this research were made possible by funding from the Australian Research Council (Grant DP16010997). The surface temperature datasets were obtained from the SCAR READER project (<https://legacy.bas.ac.uk/met/READER>). We used the observation-based Southern Hemisphere Annular Mode Index (available from <https://legacy.bas.ac.uk/met/gjma/sam.html>). To composite atmospheric and oceanic maps, we used the CFSR and ERA5 reanalyses provided by the ECMWF and NCEP. We thank Sara J. Mason, M.Sc., from Edanz Group (www.edanzediting.com/ac) for correcting a draft of this manuscript.

Author contributions

K. S. and J. I. designed the research. K.S., J. I. and I. S. wrote the paper. I. R. made cyclone tracks data sets. K. S. conducted the experiments using the LBM. All authors commented on the manuscript.

Competing interests

The authors declare no competing interests.

Fig. 1 Surface anomalies of West Antarctica. (a) Time series of averaged surface air temperature anomalies ($^{\circ}\text{C}$, deviation from climatology for 1979–2018) during June and July at the Russian Bellingshausen station. Red and blue dots indicate the top six warm winters and seven cold winters. Dashed lines show one standard deviation ($\pm 1\sigma$). (b) Difference map of sea ice concentration (%: shaded) and ice drifting speed (m/s: vector) between warm and cold winters. Dotted areas denote significant differences exceeding the 95% confidence level.

(a) T700 & SLP difference Warm - Cold

(b) Z300 difference Warm - Cold

Fig. 2 Atmospheric circulation anomalies between warm and cold winters. Difference maps for (a) air temperature at 700 hPa ($^{\circ}\text{C}$: shaded) and sea level pressure (hPa: contour), and (b) geopotential height at 300 hPa (m: shaded) between warm and cold winters. Arrows indicated horizontal component of wave-activity flux (m^2/s^2) at 300 hPa by ref. 53. Dotted areas denote significant differences exceeding the 95% confidence level.

(a) Cyclone densities difference Warm - Cold
& Cyclone density _{Clim.}

(b) Precipitation difference Warm - Cold

Fig. 3 Densities and activities of cyclones differences between warm and cold winters. Difference maps for (a) cyclone density (shaded) with climatological cyclone density (contour) and (b) precipitation (mm / month) between warm and cold winters. Contours indicate composite mean densities of cyclones (count / S) in warm winters in (a). Dotted areas denote significant differences exceeding the 95% confidence level.

Fig. 4 Oceanic and atmospheric differences over the Tasman Sea. Difference maps for SSTs between warm and cold winters ($^{\circ}\text{C}$: shaded) with mean SST for 1979 to 2018 ($^{\circ}\text{C}$: contours) in the Tasman Sea region. (b) Difference of U-wind speed (m/s: shaded) between warm and cold winters with mean U-wind (m/s: contour) for 1979 to 2018. Dotted areas indicate domains over which the differences are significant at the 95% confidence level.

Fig. 5 Atmospheric responses to heating anomalies obtained by the LBM. (a) Difference map of vertical averaged diabatic heating rate (Q1: K / day) from sigma 1.0-0.3 between warm and cold winters. (b) Mean steady response of Z300 (m) to Q1 difference over the entire globe (90°S-90°N, 0°-360°E). (c) and (d) Same as (b), but for poleward of midlatitude (90°S-30°S, 0°-360°E) in the Southern Hemisphere enclosed by purple dashed lines in (c), and the Pacific sector of the Southern Ocean (90°S-30°S, 150°-330°E) enclosed by purple dashed lines in (d), respectively. The atmospheric responses were averaged from the results obtained from the ensemble of LBM simulations undertaken with the 13 background states. **Dotted areas indicate domains over which the differences are significant at the 95% confidence level.**

Warm - Cold

La Nina - El Nino

SLP & T700 differences

Z300 & WAF300 differences

U300 difference
U300 Clim.

SST difference
SST Clim.

SAM+ - SAM-

IOD- - IOD+

SLP & T700 differences

Z300 & WAF300 differences

U300 difference
U300 Clim.

SST difference
SST Clim.

Supplementary Fig. 1 Atmospheric and ocean anomalies. Difference maps for (a) air temperature at 700 hPa ($^{\circ}\text{C}$: shaded) and sea level pressure (hPa: contour), (b) geopotential height at 300 hPa (m: shaded) with the horizontal component of wave-activity flux anomalies (m^2/s^2 : vector) at 300 hPa by ref. 53, (c) U-wind speed (m/s: shaded) at 300 hPa (U300) and (d) sea surface temperature differences between warm and cold winters. Black contours show climate values from 1979 to 2018 for (c) U300 and (d) SST. Dotted areas denote significant differences exceeding the 95% confidence level. (e-h) (i-l) (m-p) (q-t) (u-x) Same as (a-d), but for between La Niña and El Niño winters (e-h), between positive SAM and negative SAM winters (i-l), between negative IOD and positive IOD winters (m-p), between warm and cold winters without strong ENSO winters (q-t), between warm and cold winters without strong ENSO winters and (u-x) between warm and cold winters without strong SAM winters.

(a) SST [Tasman Sea] vs T700 (JUN–JUL)

(b) T700 [Antarctic Peninsula] vs SST (JUN–JUL)

Supplementary Fig. 2 Relationships between SST and T700 over the Southern Hemisphere during June and July. Map of correlation coefficients (a) between SST in the Tasman Sea (green box) and air temperature at 700 hPa, and (b) between air temperature at 700 hPa over the Antarctic Peninsula (green box) and SST. Dotted areas denote significant differences exceeding the 95% confidence level.

(a) Cyclone tracks and densities ^{Warm}
& Cyclone density _{Warm}

(b) Cyclone tracks and densities ^{Cold}
& Cyclone density _{Cold}

Supplementary Fig. 3 Cyclone density and tracks for each winter. Composite densities of cyclones (color contours) and cyclone tracks (grey lines) for (a) warm and (b) cold winters.

(a) U300 and Z300 warm winters

(b) U300 and Z300 cold winters

Supplementary Fig. 4 U-wind speed and geopotential height at 300 hPa. Composite U-wind speed (m/s: shading) and geopotential height (m: color contours) for (a) warm and (b) cold winters.

Supplementary Fig. 5 Atmospheric and oceanic fields differences over the Tasman Sea. (a) Difference maps of SST between warm and cold winters (°C: shaded) with SST for warm winters (°C: contour) in the Tasman Sea region. (b) Difference in SST as a function of latitude, averaged between 160°E and 170°E within the blue rectangle area (35°S-60°S, 160°E-170°E). (c) Difference of U-wind speed (m/s: shaded) between warm and cold winters with U-wind (m/s: contour) for warm winters as a function of latitude averaged over the blue rectangular area. Dotted areas indicate domains over which the differences are significant at the 95% confidence level.

(a) First term ($\partial\theta/\partial t$) of Q1 difference Warm - Cold

(b) Second term ($\mathbf{v} \cdot \nabla\theta$) of Q1 difference Warm - Cold

(c) Third term ($\omega \partial\theta/\partial p$) of Q1 difference Warm - Cold

Supplementary Fig. 6 Atmospheric heating anomalies for each terms of Q1. (a) Local tendency (first) term, (b) horizontal advection (second) term, and (c) vertical advection (third) term of vertically averaged diabatic heating (Q1: K / day) differences from sigma 1.0-0.3 between warm and cold winters.

Atmospheric responses for Z300

(a) Tropical regions [20°S–20°N, 0–360°E]

(b) Tasman Sea [30–90°S, 150–210°E]

(c) Amundsen Sea [30–90°S, 210–270°E]

(d) Drake Passage [30–90°S, 270–330°E]

Supplementary Fig. 7 Atmospheric responses to heating anomalies obtained by the LBM. Mean steady response of Z300 (m) to Q1 difference over the (a) Amundsen Sea area (90°S–20°N, 210°–270°E), (b) Drake Passage area (90°S–20°N, 270°–330°E), (c) Tasman Sea area (90°S–20°N, 150°–210°E) and (d) Tropical area (20°S–20°N, 0°–360°E), respectively. The atmospheric responses were averaged from the results obtained from the ensemble of LBM simulations undertaken with the 13 background states. **Dotted areas indicate domains over which the differences are significant at the 95% confidence level.**

Warm – Cold at six stations

Supplementary Fig. 8 Atmospheric and oceanic anomalies between warm and cold winters at six stations in Antarctic Peninsula. (a) Elevation map of the Antarctica Peninsula and positions of six station in AP (Bellingshausen, O'Higgins, Esperanza, Marambio, Vernadsky and Rothera). (b) Time series of averaged surface air temperature anomalies ($^{\circ}\text{C}$, deviation from climatology for 1979-2018) during June and July at six stations. Difference maps for (c) air temperature at 700 hPa ($^{\circ}\text{C}$: shaded) and sea level pressure (hPa: contour), (d) geopotential height at 300 hPa (m: shaded) with the horizontal component of wave-activity flux anomalies (m^2/s^2 : vector) at 300 hPa by ref. 53 and (e) sea surface temperature differences between warm and cold winters. Black contours show climate values from 1979 to 2018 for (e) SST. Dotted areas denote significant differences exceeding the 95% confidence level.

ERA5

(a) T700 & SLP difference Warm - Cold

(b) Z300 difference Warm - Cold

Supplementary Fig. 9 Atmospheric circulation anomalies between warm and cold winters in the ERA5. (a) Air temperature at 700 hPa (°C: shaded) and sea level pressure (hPa: contours), (b) geopotential height at 300 hPa (m: shaded) differences between warm and cold winters. Arrows indicated horizontal component of wave-activity flux anomalies (m²/s²: vector) at 300 hPa by ref. 53. Dotted areas denote significant differences exceeding the 95% confidence level. These results came from the ECMWF ERA5 dataset.

Supplementary Fig. 10 Time series of index. Time series of (a) the NINO3.4 index, (b) the SAM index, (c) IOD index during June and July. Red and blue dots indicated strong El Niño, positive SAM, positive IOD winters and strong La Niña, negative SAM, negative IOD winters. (d) Time series of averaged surface air temperature anomalies ($^{\circ}\text{C}$, deviation from climatology for 1979-2018) during June and July at the Russian Bellingshausen station. Red and blue dots indicate the warm and cold winters for which the magnitude of the temperature anomaly values exceeded one standard deviation without strong El Niño and La Niña winters. (e) Same as (d), but for without positive and negative winters. Dashed lines show one standard deviation ($\pm 1\sigma$).

Supplementary Table 1 Background states for the LBM.

	Averaged years
State 1	1980-1984
State 2	1985-1989
State 3	1990-1994
State 4	1995-1999
State 5	2000-2004
State 6	2005-2009
State 7	2010-2014
State 8	2015-2018
State 9	1980-1989
State 10	1990-1999
State 11	2000-2009
State 12	2010-2018
State 13	1979-2018

Reviewer second round comments:

Reviewer #1 (Remarks to the Author):

The authors have made a substantial improvement for the manuscript. Now I am convinced and satisfied with the authors' arguments and justifications to my previous questioning. I recommend acceptance of this manuscript to your journal.

Reviewer #2 (Remarks to the Author):

This study is only slightly improved. I still do not recommend publication in Nature Communications. I think Scientific Reports would be more appropriate so that the conversation of Tasman Sea/marine heat waves and Antarctic climate change can at least get started. But currently there are too many significant issues that prevent my recommendation of this study in its current form for publication to Nature Communications.

I do think the physical interpretation and the modeling results seem sound. Therefore Scientific Reports would be a good place. I agree with the authors that warming in the Tasman Sea could shift the storm track poleward and deepen the Amundsen Sea Low which could warm the Peninsula. This is interesting, as reviewer 1 points out, but as reviewer 3 pointed out, this has already been discussed in previous literature, and so this is not novel enough for Nature Communications in my opinion. More importantly, however, when I read the manuscript and rebuttal it just feels like the authors are showing bits and pieces of analysis to support the current conclusions and I don't feel like the authors are digging deep to see if their findings are robust in space and time and the authors are avoiding addressing the glaring issues such as their use of one station and two months. Further, it was my impression that the Tasman Sea heat waves have been mostly in the summer (not winter) and also mostly in the 21st century, not the 20th century when the warming on the Peninsula stopped. This is another major issue that the authors do not address.

If this study were to be published in Nature Communications, I would expect to see a robust relationship that exists in space (for example, for more than one station) and time (for example, for more than just two months) along with a clear explanation as to why the connection is not seen at other places/times. All of this is missing, as when looking at the six Peninsula stations, a strong La Nina connection is still present which is known to shift the storm track poleward, and when the analysis includes August, the relationship with the Tasman Sea disappears. Therefore the connection is not robust. Maybe this can all be due to a striking lack of detail, transparency, and justification for many choices and statements used throughout the study, I'm not sure.

Major comments

Your justification for using Bellingshausen station and June and July are still not sufficient or appropriate. Though you never state the time period of your study in the manuscript (this needs added), I am guessing it is 1979-2018. The Peninsula station temperature data is all pretty well complete after 1979, so data availability is not an issue. Further, looking at 20th century temperature trends over common period for the stations with long records (1970-2000), Vernadsky and Marambio had stronger warming during 20th century than Bellingshausen (I didn't include Rothera because it begins in late 1970s and I didn't look at O'Higgins). Therefore the authors aren't even focusing on a station that is best representative of the strong 20th century Peninsula warming, yet this is a large basis of the study. Looking at all six stations in supplementary Fig. 8, the central tropical Pacific SSTs look very important as I commented on previously, and there is significant wave flux coming out of the central tropical Pacific indicating that indeed ENSO is important. Therefore I suspect that Bellingshausen station is the only station with a relationship with Tasman Sea SST but authors are not being transparent about this. And given that when adding August to the analysis in response Fig A6 you also lose the Tasman Sea connection, then the relationship is not robust in time...why?

Why don't you see the relationship in other months/season or for other parts of the Peninsula? This needs to be addressed.

Another major concern is the discussion on marine heat waves to justify looking at the Tasman Sea. To my knowledge (I definitely may be wrong) these heat waves have mostly been in the summer and mostly during the 21st century. This study focuses on the winter and a warming trend on Peninsula that occurred during the 20th century. There is a major mismatch here that needs addressed and explained.

Minor comments

How you calculated wave flux needs to be added to the Methods.

How you calculate significance needs added to the Methods.

How did you choose which ENSO and SAM events to not include in your supplementary Fig. 1? Your threshold and reasoning needs added to the Methods.

Please change all instances of Amundsen-Bellingshausen Seas Low to Amundsen Sea Low to be consistent with recent literature.

Please refer to the Peninsula warming in past tense, not present tense.

There are a significant number of typos and grammatical mistakes throughout the manuscript that need fixed.

Reviewer #3 (Remarks to the Author):

First, I would like to thank the authors for correcting most of my previous concerns. However, I still have one major one and one minor one left.

The major issue is the point regarding using the Bellingshausen data to represent the Antarctic Peninsula as a whole, which was also picked up on by Referee #2. I would argue that some of the authors' justifications for doing so are spurious and that they should indeed use the combination of the six stations as suggested.

(i) If the results of using Bellingshausen and the combined six stations are very similar, as the authors state, then all the justifications they talk about are irrelevant. A 'regional' temperature should be based on temperatures from across the region, not a single point.

(ii) All stations have temperature variability caused by local effects. For example, if the authors compare the READER data from Bellingshausen and Marsh, which is located literally 200 m away, they will see that monthly mean temperatures can disagree by as much as 0.5°C.

(iii) There is only one summer when there is insufficient data from Rothera to provide a JJ temperature value (1999).

(iv) There is a very easy way to assess the quality of surface data at Esperanza, by comparing the CLIMAT values for those months where there are also SYNOP data to see if there is any bias. For example, a quick comparison of ten years of June data from 2008-2017 indicates a mean difference of 0.1°C, so probably of similar magnitude to the error in the thermometer being used to measure the temperature.

(v) Foehn winds are much more prevalent on the eastern side of the Peninsula (e.g. Esperanza and Marambio) than on the west (Vernadsky and Rothera). The only Kutita et al. (2016) paper I could locate was entitled 'Influence of large-scale atmospheric circulation on marine air intrusion toward the East Antarctic coast' and I could find no reference to the Antarctic Peninsula in it.

The minor issue concerns the new references, which have been formatted incorrectly in many cases. In particular, for those authors where there are more than one initial I think the order of the initials is usually incorrect and indeed is inconsistent across the reference list

Response to reviewer #1: Sato et al. (NCOMMS-19-32930-A)

Reviewer #1 (Remarks to the Author):

The authors have made a substantial improvement for the manuscript. Now I am convinced and satisfied with the authors' arguments and justifications to my previous questioning. I recommend acceptance of this manuscript to your journal.

We very much appreciate your very positive evaluation of our revision, and your 'accept' recommendation.

Response to reviewer #2: Sato et al. (NCOMMS-19-32930-A)

We would like to thank you for providing constructive comments regarding our manuscript. The point-by-point responses are made below.

Reviewer #2 (Remarks to the Author):

This study is only slightly improved. I still do not recommend publication in Nature Communications. I think Scientific Reports would be more appropriate so that the conversation of Tasman Sea/marine heat waves and Antarctic climate change can at least get started. But currently there are too many significant issues that prevent my recommendation of this study in its current form for publication to Nature Communications.

I do think the physical interpretation and the modeling results seem sound. Therefore Scientific Reports would be a good place. I agree with the authors that warming in the Tasman Sea could shift the storm track poleward and deepen the Amundsen Sea Low which could warm the Peninsula. This is interesting, as reviewer 1 points out, but as reviewer 3 pointed out, this has already been discussed in previous literature, and so this is not novel enough for Nature Communications in my opinion. More importantly, however, when I read the manuscript and rebuttal it just feels like the authors are showing bits and pieces of analysis to support the current conclusions and I don't feel like the authors are digging deep to see if their findings are robust in space and time and the authors are avoiding addressing the glaring issues such as their use of one station and two months. Further, it was my impression that the Tasman Sea heat waves have been mostly in the summer (not winter) and also mostly in the 21st century, not the 20th century when the warming on the Peninsula stopped. This is another major issue that the authors do not address.

If this study were to be published in Nature Communications, I would expect to see a robust relationship that exists in space (for example, for more than one station) and time (for example, for more than just two months) along with a clear explanation as to why the connection is not seen at other places/times. All of this is missing, as when looking at the six Peninsula stations, a strong La Nina connection is still present which is known to shift the storm track poleward, and when the analysis includes August, the relationship with the Tasman Sea disappears. Therefore the connection is not robust. Maybe this can all be due to a striking lack of detail, transparency, and justification for many choices and statements used throughout the study, I'm not sure.

We appreciate the reviewer's suggestion of an alternative journal for our paper. However, in undertaking the present research and planning the structure of the paper we were very aware of the scope of *Nature Communications* in '... publishing high-quality research in all areas of the biological, physical, chemical and Earth sciences. Papers published by the journal represent important advances of significance to specialists within each field'. Our submission fits perfectly in with these guidelines.

Also, thank you for drawing attention to the very relevant comments of reviewer #3, as this clarifies a number of issues here. In their assessment of the **original** submission reviewer #3 had commented

...

'A lot of the analysis of atmospheric changes associated with warm Antarctic Peninsula winters follows a very similar methodology to a paper written more than 20 years ago

(Marshall & King 1998 in GRL). **While the specific influence of Tasman Sea temperatures in contributing to these changes and the associated model experiments are indeed new** [our emphasis, and indeed we note that no previous study reported the impact of SST anomaly over *any* part of the SH midlatitudes on these atmospheric circulations anomalies], I would have liked to see a greater discussion of how this finding related to other influences on Antarctic climate (other than ENSO and the SAM).

In commenting on our revised submission reviewer #3 said ...

‘First, I would like to thank the authors for correcting most of my previous concerns. However, I still have one major one and one minor one left. [That major point only concerns the need for six stations in the analysis (instead of the single station of Bellingshausen). This has been done in the present submission. Reviewer #3 now makes no mention of the Tasman Sea temperatures (and was clearly satisfied with how his/her original comments were addressed).]

The major issue is the point regarding using the Bellingshausen data to represent the Antarctic Peninsula as a whole, which was also picked up on by Referee #2. I would argue that some of the authors’ justifications for doing so are spurious and that they should indeed use the combination of the six stations as suggested.’

Reviewer #3 highlights here only the need for six stations in the analysis (instead of the single station of Bellingshausen). This has been done in the present submission. Reviewer #3 now makes no mention of the Tasman Sea temperatures (and was clearly satisfied with how his/her original comments were addressed).

All the other points raised in the second paragraph of Reviewer #2 (above) are raised again below. We address them there.

With regard to the statement of reviewer #2 that ...

‘ ... I don't feel like the authors are digging deep to see if their findings are robust in space and time and the authors are avoiding addressing the glaring issues such as their use of one station and two months’

We accept the wisdom of looking at the six Peninsula stations during three winter months. This has now been incorporated into our study.

Reviewer #3 commented on our revised submission ...

‘Further, it was my impression that the Tasman Sea heat waves have been mostly in the summer (not winter) and also mostly in the 21st century, not the 20th century when the warming on the Peninsula stopped. This is another major issue that the authors do not

address'

Here, reviewer #3 mentioned the *trends* in Tasman Sea SST and Antarctic Peninsula temperatures. However, our study is focused on the ***interannual connection*** between Tasman Sea SST and Antarctic Peninsula temperatures, rather than what the *trends* in these parameters might be (the 'fast' and 'slow' physics between these will be quite different - e.g., involvement or otherwise of the middle or deep ocean).

In the third paragraph, reviewer #3 mentioned ...

'All of this is missing, as when looking at the six Peninsula stations, a strong La Nina connection is still present which is known to shift the storm track poleward, and when the analysis includes August, the relationship with the Tasman Sea disappears'

Present study, which based on averaged temperature anomalies for three winter months at six Peninsula stations, reveals that significant warming over the Tasman Sea leads to poleward shift of cyclone tracks, even without tropical Pacific cooling. We raised these points again in response for major comments of reviewer #2.

Major comments

Your justification for using Bellingshausen station and June and July are still not sufficient or appropriate. Though you never state the time period of your study in the manuscript (this needs added), I am guessing it is 1979-2018. The Peninsula station temperature data is all pretty well complete after 1979, so data availability is not an issue. Further, looking at 20th century temperature trends over common period for the stations with long records (1970-2000), Vernadsky and Marambio had stronger warming during 20th century than Bellingshausen (I didn't include Rothera because it begins in late 1970s and I didn't look at O'Higgins). Therefore the authors aren't even focusing on a station that is best representative of the strong 20th century Peninsula warming, yet this is a large basis of the study. Looking at all six stations in supplementary Fig. 8, the central tropical Pacific SSTs look very important as I commented on previously, and there is significant wave flux coming out of the central tropical Pacific indicating that indeed ENSO is important. Therefore I suspect that Bellingshausen station is the only station with a relationship with Tasman Sea SST but authors are not being transparent about this. And given that when adding August to the analysis in response Fig A6 you also lose the Tasman Sea connection, then the relationship is not robust in time...why?

Thank you for this point. In revised article, we conducted similar analyses based on averaged surface air temperature anomalies at six stations for three winter months. In warm three winter months at six stations, the atmospheric (e.g. Antarctic Peninsula warming, the Amundsen Seas Low deepen) and oceanic (e.g. Tasman warming) anomaly patterns are similar to those in warm two winter months at only Bellingshausen station (Figs. 1, 2 and Figs. S1-S4, S6, S13). When strong ENSO winters were removed from warm Antarctic Peninsula winters, these atmospheric and oceanic anomalies remain strong, even absent of tropical Pacific SST cooling (Fig. 3 and Figs. S5-S7). In addition, to assess the impacts of SST anomalies over the tropical Pacific region and Tasman Sea on atmospheric circulations in the Southern Hemisphere, we calculated atmospheric responses to SST anomalies in warm winters using the atmospheric general circulation model (Fig. 4 and Figs. S8,9). From these results, we conclude that although the tropical SST cooling associated with ENSO has an important role in the Antarctic Peninsula warming, the Tasman Sea SST anomalies alone produce warm winters in the Antarctic Peninsula.

We revised the sentences (lines 77-98, 100-132, 134-160 and 162-207) and figures (Fig. 1-3 and Fig S1-S7, S13).

Why don't you see the relationship in other months/season or for other parts of the Peninsula? This needs to be addressed.

Many thanks for this good suggestion, which we have now acted upon. The sentences (lines 263-284, lines 296-299) and figures (Figs. S11, 12) about these results for another three seasons are added to revised manuscript.

Another major concern is the discussion on marine heat waves to justify looking at the Tasman Sea. To my knowledge (I definitely may be wrong) these heat waves have mostly been in the summer and mostly during the 21st century. This study focuses on the winter and a warming trend on Peninsula that occurred during the 20th century. There is a major mismatch here that needs addressed and explained.

Shears and Bowen (2017) suggested that the Tasman Sea has warming anomalies and trends in all seasons over the period 1967 to 2016. However, as stressed above, our study is focused on **interannual associations** between Tasman Sea SST and Antarctic Peninsula temperatures, not linkage in the trends between these parameters. We added the following sentences in lines 84-87.

“Although weak negative trends of AP air temperature since 1999 were reported by a previous study¹, our analysis shows that the AP stations experienced eight warm winters since that time (in 2000, 2004, 2008, 2010, 2014, 2016, 2018 and 2019).”

and (lines 144-146)

“Therefore, these atmospheric circulation changes related to anomalous SST warming over the Tasman Sea would contribute to recent anomalous warm AP winters.”

Minor comments

How you calculated wave flux needs to be added to the Methods.

Details of wave activity flux are presented in the paper of Takaya and Nakamura (2001). Therefore, we added the following sentence to Methods (lines 306-307):

“The wave activity flux representing stationary Rossby wave propagation was based on ref. 55.”

How you calculate significance needs added to the Methods.

The sentence is added below to Methods (lines 307-309):

“In this study, the significance test was a standard two-tailed *t*-test with degrees of freedom based on the number of years in the figures and supplementary figures.”

How did you choose which ENSO and SAM events to not include in your supplementary Fig. 1? Your threshold and reasoning needs added to the Methods.

From NINO3.4 and SAM index anomalies, we determined strong ENSO and SAM winters. Strong ENSO and SAM winters for which these anomalies values exceeded one half standard deviation. We added relevant sentences to Method.

Please change all instances of Amundsen-Bellinghausen Seas Low to Amundsen Sea Low to be consistent with recent literature.

Thanks. We have adopted your suggestion.

Please refer to the Peninsula warming in past tense, not present tense.

We added the sentence below (lines 84-87):

“Although weak negative trends of AP air temperature since 1999 were reported by a previous study¹, our analysis shows that the AP stations experienced eight warm winters since that time (in 2000, 2004, 2008, 2010, 2014, 2016, 2018 and 2019)”

There are a significant number of typos and grammatical mistakes throughout the manuscript that need fixed.

Thank you. The text has now undergone rigorous editorial examination and correction.

Response to reviewer #3: Sato et al. (NCOMMS-19-32930-A)

First, I would like to thank the authors for correcting most of my previous concerns. However, I still have one major one and one minor one left.

The major issue is the point regarding using the Bellingshausen data to represent the Antarctic Peninsula as a whole, which was also picked up on by Referee #2. I would argue that some of the authors' justifications for doing so are spurious and that they should indeed use the combination of the six stations as suggested.

(i) If the results of using Bellingshausen and the combined six stations are very similar, as the authors state, then all the justifications they talk about are irrelevant. A 'regional' temperature should be based on temperatures from across the region, not a single point.

(ii) All stations have temperature variability caused by local effects. For example, if the authors compare the READER data from Bellingshausen and Marsh, which is located literally 200 m away, they will see that monthly mean temperatures can disagree by as much as 0.5°C.

(iii) There is only one summer when there is insufficient data from Rothera to provide a JJ temperature value (1999).

(iv) There is a very easy way to assess the quality of surface data at Esperanza, by comparing the CLIMAT values for those months where there are also SYNOP data to see if there is any bias. For example, a quick comparison of ten years of June data from 2008-2017 indicates a mean difference of 0.1°C, so probably of similar magnitude to the error in the thermometer being used to measure the temperature.

(v) Foehn winds are much more prevalent on the eastern side of the Peninsula (e.g. Esperanza and Marambio) than on the west (Vernadsky and Rothera). The only Kutita et al. (2016) paper I could locate was entitled 'Influence of large-scale atmospheric circulation on marine air intrusion toward the East Antarctic coast' and I could find no reference to the Antarctic Peninsula in it.

Thank you very much for your encouragement.

(i) As indicated above, we fully accept the value of using the six-station average, and our investigation is now conducted with these temperatures. Using this spatially-averaged temperature we obtained similar atmospheric (e.g. Antarctic Peninsula warming, the Amundsen Seas Low deepen) and oceanic (e.g. Tasman warming) anomaly patterns in warm three winter month to those found previously in warm two winter months at only Bellingshausen station (Figs. 1-2 and Figs. S1-S4, S6, S13)

(ii) – (v) Thanks for these additional insights. Our revised analysis is based on temperature anomalies at six stations, so the points raised here are no longer issues.

The minor issue concerns the new references, which have been formatted incorrectly in many cases. In particular, for those authors where there are more than one initial I think the order of the initials is usually incorrect and indeed is inconsistent across the reference list.

Thank you very much for spotting those. The relevant references have now been corrected.

Reference

Shears, N. T., and M. M. Bowen, 2017: Half a century of coastal temperature records reveal complex warming trends in western boundary currents. *Scientific Reports*, **7**, 14527, doi: 10.1038/s41598-017-14944-2.

Reviewer comments, third round-

Reviewer #3 (Remarks to the Author):

Thank you for your dedicated work on this manuscript. The reviewer admires your passion and perseverance and has no further comments to be addressed.

Karen Bandeen-Roche

Reviewer third round comments:

Reviewer #2 (Remarks to the Author):

Overall this study is much improved from previous versions and I commend the authors for the great effort. It is now much clearer what the relative roles of ENSO, SAM, and Tasman Sea are, and I am now convinced Tasman Sea warming contributes to AP warming in winter (and potentially spring and autumn) both in addition to SAM/ENSO and independently of SAM/ENSO. This study now serves as an excellent contribution to our understanding of West Antarctic and AP climate. I'm happy to recommend for publication with minor revisions outlined below.

Specific comments

Why does the first sentence of the Introduction focus on absence of West Antarctic warming during the summer season? This is confusing. Isn't it more appropriate and relevant to begin the paper talking about the AP temperature trends? I.e., please be specific about the temporal and spatial variability of the AP temperature trends (seasonality, decadal variability, and west vs. east AP). The current wording of this paragraph is misleading... makes it seem like natural variability has caused absence of summer warming in West Antarctica, which isn't necessarily true (if would likely be tied to ozone depletion, and if my memory is correct there was never very strong or statistically significant summer warming of West Antarctica), while the AP has been warming unabated. This is misleading. West Antarctic, west AP, and east AP temperature trends have all exhibited their own marked seasonal and decadal variability which should be made clearer in this opening paragraph.

L31: international -> internal

L57-60: "Numerical experiments have shown that the heating anomalies over the tropical region..." Heating anomalies where?

L72: mid-latitudes -> Southern Hemisphere mid-latitudes

L82: "Thirteen years" and "12 winters" ... are these both for winter? Please use consistent terminology.

L92-93: Or, the warming caused the sea ice loss?

L107-108: Please reword this sentence, it's a bit confusing.

L155-156: Couldn't the Tasman Sea warming be a result of the poleward shift of the polar-front jet due to La Nina/+SAM, rather than the Tasman Sea warming causing the poleward shift of the jet? Your analysis doesn't rule this out.

Reviewer #3 (Remarks to the Author):

I thank the authors for dealing with my major concern regarding the number of stations used to compute a regional climate signal for the Antarctic Peninsula.

However, I notice there are still many issues with the references regarding the order of initials despite the authors stating that 'The text has now undergone rigorous editorial examination and correction'. In many cases there is inconsistency between the same author: e.g. Ryan Fogt in references 14 and 21. While we all make mistakes, to me this suggests a lack of care and indeed rigour by the authors in dealing with the revision process.

I would also like to state that just because I haven't pointed something out in the manuscript (possible issues with the Tasman Sea temperatures) that does not justify using that as an argument against the concerns of another reviewer who may well know more about this particular subject than I do.

We are very grateful for these very helpful additional comments by Reviewers #2 and #3. The point-by-point responses are made below.

Response to reviewer #2: Sato et al. (NCOMMS-19-32930B-Z)

Reviewer #2 (Remarks to the Author):

Overall this study is much improved from previous versions and I commend the authors for the great effort. It is now much clearer what the relative roles of ENSO, SAM, and Tasman Sea are, and I am now convinced Tasman Sea warming contributes to AP warming in winter (and potentially spring and autumn) both in addition to SAM/ENSO and independently of SAM/ENSO. This study now serves as an excellent contribution to our understanding of West Antarctic and AP climate. I'm happy to recommend for publication with minor revisions outlined below.

Specific comments

Why does the first sentence of the Introduction focus on absence of West Antarctic warming during the summer season? This is confusing. Isn't it more appropriate and relevant to begin the paper talking about the AP temperature trends? I.e., please be specific about the temporal and spatial variability of the AP temperature trends (seasonality, decadal variability, and west vs. east AP). The current wording of this paragraph is misleading... makes it seem like natural variability has caused absence of summer warming in West Antarctica, which isn't necessarily true (if would likely be tied to ozone depletion, and if my memory is correct there was never very strong or statistically significant summer warming of West Antarctica), while the AP has been warming unabated. This is misleading. West Antarctic, west AP, and east AP temperature trends have all exhibited their own marked seasonal and decadal variability which should be made clearer in this opening paragraph.

We very much appreciate your very positive evaluation of our revision, and your 'accept' recommendation. Following your good suggestion we now make appropriate mention of the temporal and spatial variability of AP temperature trends in the opening paragraph in new version of manuscript (lines 32-42).

L31: international -> internal

Thanks for spotting this. The appropriate correction has been made.

L57-60: "Numerical experiments have shown that the heating anomalies over the tropical region..." Heating anomalies where?

We apologize that the wording of this sentence was not at all clear. The sentence has now been changed to ... (lines 61-63)

'Numerical experiments have shown that warming over the tropical Atlantic and Indian Oceans and cooling over the tropical eastern Pacific Ocean result in a deeper ASL and West Antarctic warming²³.'

L72: mid-latitudes -> Southern Hemisphere mid-latitudes

To avoid any ambiguity we have inserted the suggested 'Southern Hemisphere'.

L82: "Thirteen years" and "12 winters" ... are these both for winter? Please use consistent terminology.

We changed "12" to "twelve".

L92-93: Or, the warming caused the sea ice loss?

The sentences have been revised to reflect the possible complexity (lines 100-105).

L107-108: Please reword this sentence, it's a bit confusing.

We agree. The sentence has now been revised (lines 115-116).

L155-156: Couldn't the Tasman Sea warming be a result of the poleward shift of the polar-front jet due to La Nina/+SAM, rather than the Tasman Sea warming causing the poleward shift of the jet? Your analysis doesn't rule this out?

We have changed the sentence to reflect this possible interpretation (lines 164-167).

Response to reviewer #3: Sato et al. (NCOMMS-19-32930B-Z)

Reviewer #3 (Remarks to the Author):

I thank the authors for dealing with my major concern regarding the number of stations used to compute a regional climate signal for the Antarctic Peninsula.

However, I notice there are still many issues with the references regarding the order of initials despite the authors stating that 'The text has now undergone rigorous editorial examination and correction'. In many cases there is inconsistency between the same author: e.g. Ryan Fogt in references 14 and 21. While we all make mistakes, to me this suggests a lack of care and indeed rigour by the authors in dealing with the revision process.

I would also like to state that just because I haven't pointed something out in the manuscript (possible issues with the Tasman Sea temperatures) that does not justify using that as an argument against the concerns of another reviewer who may well know more about this particular subject than I do.

We very much appreciate your very positive evaluation of our revision. We do agree that we should have been much more careful with regard to the editorial issue that the reviewer raises. Our revision now strictly adheres to the *Nature Communications* referencing style.